# Shallow Cumuli Cover and Its Uncertainties from Ground-based Lidar-Radar Data and Sky Images

Erin A. Riley[1], Jessica M. Kleiss[1,*], Laura D. Riihimaki[2,3], Charles N. Long[2,3], Larry K. Berg[4] and Evgueni Kassianov[4,*]

[1] Environmental Studies, Lewis and Clark College, Portland, OR 97219, USA

[2] Cooperative Institute for Research in Environmental Sciences, University of Colorado, Boulder, CO 80309, USA

[3] Global Monitoring Division, NOAA Earth System Research Laboratory, Boulder, CO, 80305, USA

[4] Pacific Northwest National Laboratory, Richland, WA 99354, USA

*Correspondence to*: Jessica M. Kleiss, jkleiss@lclark.edu, Evgueni Kassianov, Evgueni.Kassianov@pnnl.gov

**Abstract**

Cloud cover estimates of single-layer shallow cumuli obtained from narrow field-of-view (FOV) lidar-radar and wide-FOV Total Sky Imager (TSI) data are compared over an extended period (2000-2017 summers) at the established United States Atmospheric Radiation Measurement mid-continental Southern Great Plains site. We quantify the impacts of two factors on hourly and sub-hourly cloud cover estimates: 1) instrument-dependent cloud detection and data merging criteria, and 2) FOV configuration. Enhanced observations at this site combine the advantages of the ceilometer, micropulse lidar (MPL) and cloud radar in merged data products. Data collected by these three instruments are used to calculate narrow-FOV cloud fraction (CF) as a temporal fraction of cloudy returns within a given period. Sky images provided by TSI are used to calculate the wide-FOV fractional sky cover (FSC) as a fraction of cloudy pixels within a given image. To assess the impact of the first factor on CF obtained from the merged data products, we consider two additional sub-periods (2000-2010 and 2011-2017 summers) that mark significant instrumentation and algorithmic advances in the cloud detection and data merging. We demonstrate that CF obtained from ceilometer data alone and FSC obtained from sky images provide the most similar and consistent cloud cover estimates: hourly bias and root-mean-square difference (RMSD) are within 0.04 and 0.12, respectively. Whereas CF from merged MPL-ceilometer data provides the largest estimates of the multi-year mean cloud cover: about 0.12 (35%) and 0.08 (24%) greater than FSC for the first and second sub-periods, respectively. CF from merged ceilometer-MPL-radar data has the strongest sub-period dependence with a bias of 0.08 (24%) compared to FSC for the first sub-period and shows no bias for the second sub-period. The strong period dependence of CF obtained from the combined ceilometer-MPL-radar data is likely results from a change in what sensors are relied on to detect clouds below 3 km. After 2011, the MPL stopped being used for cloud top height detection below 3 km, leaving the radar as the only sensor used in clouds top height retrievals. To quantify the FOV impact, a narrow-FOV FSC is derived from the TSI images. We demonstrate that FOV configuration does not modify the bias, but impacts the RMSD (0.1 hourly, 0.15 sub-hourly). In particular, the FOV impact is significant for sub-hourly observations, where 41% of narrow- and wide-FOV FSC differ by more than 0.1. A new "quick-look" tool is introduced

to visualize impacts of these two factors through integration of CF and FSC data with novel TSI-based images of the spatial variability in cloud cover. The influence of cloud field organization, such cloud streets parallel to the wind direction, on narrow- and wide- FOV cloud cover estimates can be visually assessed.

## 1 Introduction

Shallow cumuli (ShCu) have a number of important roles in the Earth's climate system due to their complex interactions with radiation, the atmosphere and the surface (e.g., Arakawa, 2004; Vial et al., 2017; Park and Kwon, 2018). For example, the amount of surface moisture can influence the cloud properties by altering the humidity of the boundary layer and by modifying the partitioning of the sensible and latent heat fluxes (Zhang and Klein 2013; Berg et al., 2013), while the presence of ShCu provides a negative feedback by shading the surface and reducing its solar heating (Berg et al., 2011; Xiao et al., 2018). The ShCu exhibit strong spatial and temporal variability that has been a topic of increasing interest in recent years for both observational (Berg and Kassianov, 2008; Chandra et al., 2013) and model (Yun et al., 2017; Angevine et al., 2018) studies. To improve our understanding of cloud cover variability and its impact on the intricate cloud-atmosphere-surface interactions, both long-term and detailed diurnal observations of the ShCu properties are highly desirable. There are two conventional measurement-based estimates of cloud cover: (1) cloud fraction (CF) obtained from zenith-pointing narrow-FOV observations and defined as the fraction of time when a cloud is detected within a specified period, and (2) fractional sky coverage (FSC) obtained from wide-FOV observations and defined as the fraction of cloudy pixels in a sky image. Note that FSC is similar to that estimated by a cloudy-sky observer (e.g., Henderson-Sellers and McGuffie, 1990; Kassianov et al., 2005; Long et al. 2006).

Long-term measurement-based statistics of ShCu have been employed for model-observation comparison (Zhang et al., 2017; Endo et al., 2018). The observational studies take advantage of the synergistic use of ground-based observations from the ceilometer, micropulse lidar (MPL) and millimeter cloud radar at the Atmospheric Radiation Measurement (ARM) mid-continental Southern Great Plains (SGP) site to obtain vertically-resolved hydrometeor layers with high temporal resolution (https://www.arm.gov/). A merged lidar-radar data product is available from Nov. 1996 to the present at the SGP site and has served as a basis for developing ShCu climatology at the SGP site (Berg and Kassianov 2008; Zhang and Klein 2013) and observational testbeds for model evaluation (Zhang et al., 2017).

Efforts in improving both ground-based observations and modeling of continental ShCu at the SGP site are currently underway (e.g., Gustafson et al., 2017; Zhang et al., 2017). Zhang et al., (2017) suggested that, on average, the modeled areal cloud cover tends to underestimate observed CF substantially (up to 0.1 or 65% for clouds with vertical extent greater than 0.3 km). Although several potential factors for the obtained model-data discrepancy have been considered, including different

model setups and selection of events with different vertical cloud extents, the role of observational uncertainties has not been directly addressed. For example, FSC obtained from the wide-field-of-view (FOV) Total Sky Imager (TSI) data as the fraction of cloudy pixels in a hemispherical image provides a better agreement with model outputs than the narrow-FOV CF from merged ceilometer-MPL data (Gustafson et al., 2018), indicating a potential consequence of the FOV configuration. In addition, long-term averages of CF obtained from merged ceilometer-MPL data tend to be larger than FSC acquired from collocated TSI observations (Boers et al., 2010; Qian et al., 2012; Wu et al., 2014; Kennedy et al., 2014), indicating a potential consequence of instrument-dependent cloud detection differences. Moreover, sampling of LES-generated cloud fields by a virtual instrument can be a helpful way to reconcile debated differences between the retrieved and predicted values of cloud cover (Oue et al., 2016).

It is expected that the impacts of instrument-dependent cloud detection and FOV configurations are entangled, and our study aims to assess their relative importance on cloud cover estimates of single-layer continental ShCu observed at the SGP site. In 2010, upgrades of three instruments (TSI, MPL and radar) took place, and data merging improvements were implemented on the ceilometer-MPL and ceilometer-MPL-radar (hereafter lidar-radar) merged datasets. Our study uses the natural separation provided by these upgrades and improvements to assess the relative impact of instrument-dependent cloud detection on the ShCu cover derived from three commonly used CF calculations, and FSC. Our study also uses wide-FOV and narrow-FOV TSI-based observations to assess the relative impact of the FOV configuration on FSC. Two main questions guide our investigation: 1) Have significant changes in the observations of ShCu cover occurred at the SGP site due to instrumental and algorithmic upgrades? And, 2) what is the impact of FOV configurations on hourly and sub-hourly observations of ShCu cover? These questions are addressed using statistical approaches as well as a new tool that integrates the narrow- and wide-FOV observations to visualize the two considered impacts for a given day of interest at user-specified temporal and spatial scales.

**2 Data**

Eighteen years (2000-2017) summertime (May-Sept) observations of FSC, cloud base height (CBH), cloud top height, and wind speed and direction were collected at the ARM SGP Central Facility. The data described below are available at https://www.arm.gov/data, and references to detailed instrument descriptions and records are in Table A1. Here is a short summary of data used in our analysis, and Appendix A contains pertinent information for their application.

1) **Merged lidar and lidar-radar data products**. A time-series of CBH and cloud top height values are obtained from the combined 34.86 GHz millimeter cloud radar (MMCR), MPL, and ceilometer data in the Active Remotely-Sensed Cloud Locations (ARSCL) value added product (Data reference: Johnson and Giangrande, 1996). The merged dataset improves upon the MMCR vertical profiling of hydrometeors by employing an MPL cloud mask to identify CBH and aide in mitigating the significant impact of insect returns (Clothiaux et al., 2000). The screened high temporal (10 s) and vertical (45 m) radar reflectivity best estimate is used to identify the vertical distribution of

hydrometeor layers above the SGP. The CBH best estimate is obtained from merged ceilometer-MPL data. A cloud top height estimate is derived from merging the radar reflectivity best estimate, CBH best estimate, and MPL cloud mask. In 2011 the MMCR was replaced and renamed to Ka-band Zenith Radar (KAZR) (Kollias et al. 2016) and ARSCL was subsequently updated and renamed to KAZRARSCL (Data reference: Johnson et al., 2011). A summary of the pertinent instrumental and algorithmic differences to the results herein for the periods before and after 2011 are provided in Appendix A.

2) **Ceilometer.** A time series of CBH values is obtained from the ceilometer (Data Reference: Ermold and Morris, 1997). The ceilometer records up to three cloud bases at heights up to a maximum of 7.7 km and 16 s resolution. Only the lowest cloud-base is used here.

3) **Total Sky Imager cloud mask**. In addition to the sky image, the TSI processing produces a cloud mask (Data reference: Holben et al., 1994) every 30 seconds that classifies each image pixel as clear, opaque, or thin cloud, and identifies a 25° FOV circumsolar region and an inner zenith region (Long et al., 2006). The percent error of in FSC retrieval is estimated to be 10% for 95% of observations (Long et al., 2001).

4) **ShCu Event Selection**. The newly released ARM Shallow Cumulus data product (Data Reference: Shi et al., 2000) identifies times of ShCu from lidar / radar cloud boundary heights and includes FSC from TSI observations (Lim et al., 2018). The dataset provides hourly flags for the presence of ShCu, and a flag indicating the additional presence of other cloud types, such as commonly overlying cirrus that can either be intermittent or persistent throughout the day. From the Shallow Cumulus data product, we initially select days with single-layer (no upper-level) ShCu with at least 2-h duration. Once selected, all hours in this day with ShCu are included, additionally extending the start and end-times by 1 hour each. The extension allows for more accurate determination of the start- and end- times of the event on the finer time scale of the TSI FSC (15 min). Quality control procedures (Sect. 3.4) are used to censor multi-layer clouds and clear sky conditions on the 15-min and hourly observations of cloud cover. The initial selection provided 614 candidate days with a total of 2393 hours.

5) **915 MHz Radar Wind Profiler**. Wind data are obtained at CBH from the low power setting of the 915 MHz Radar Wind Profiler which has a range-gate spacing of 60 m and hourly reporting intervals (Data reference: Muradyan and Coulter, 1997). In the event of missing data at CBH, the first available reading above 500 m is used. In the event of missing data, 10m surface wind data are used as a tertiary means to impute wind data (Data reference: ARM, 1994). The wind data is used qualitatively in this analysis.

## 3 Methods

We calculate five estimates of the cloud cover from active and passive observations. Three estimates define narrow-FOV cloud fractions (CFs) retrieved from the lidar-radar observations (Sect. 3.1) to assess the impact of the instrument-dependent cloud detection and data merging on cloud cover estimates (Sect.4.1). Two estimates define fractional sky cover (FSC) with 100°-

FOV (Sect. 3.2) and narrow-FOV (3x3 pixels within 100°FOV) (Sect.3.3) obtained from TSI observations to estimate the impact of the FOV configuration on CF and FSC comparisons (Sect. 4.2). The resulting five ShCu cloud cover estimates are then combined, and data selection and quality control procedures are applied to the dataset (Sect. 3.4). The cloud cover estimates are compared for the entire period (2000-2017) and two sub-periods (2000-2010, and 2011-2017) separated by instrumental and algorithmic upgrades (Sect.4). Finally, a heuristic tool is developed to visualize impacts of the cloud detection, data merging and FOV configuration on the cloud cover estimates (Sect. 4.3).

### 3.1. Narrow-FOV Cloud Fraction (CF)

We consider CF calculated directly from the ARSCL (or KAZRARSCL) data products as a fraction of 10 s (or 4 s) cloudy returns over a given time (e.g., Xi et al., 2010). Thus, the CF defines the frequency of cloud occurrence from the combined narrow-FOV lidar-radar measurements. ShCu returns are defined using thresholds on 10 s (or 4 s) CBH and cloud top height information (Table 1) in a binary fashion. The three estimates of ShCu cover from lidar-radar observations are 1) CF from the merged lidar-radar products uses CBH information from the merged ceilometer-MPL and cloud top height from the merged MPL-radar. This method has the advantages of low missing data due to use of multiple instruments and incorporates information about cloud top height consistent with the definition of shallow convection. The MPL has the potential to attenuate in thick clouds, necessitating radar data for cloud top height retrieval. Insect contamination may contribute to significant uncertainty of the radar-based retrievals of cloud boundaries, however, we do not expect this to significantly impact our results for several reasons. First, when using cloud top height in cloud fraction, we still require cloud base to be identified by the ceilometer or MPL, so we will not misclassify insect-only layers as cloud. Second, our results are not very sensitive to the actual value of cloud top height as long as it is below 4 km, and as most insects will be found in the boundary layer or immediately above it (Kollias et al. 2016; Wainwright et al. 2017) they are not likely to cause the radar to misidentify a cloud height above 4 km if a cloud doesn't exist at that height. Finally, both the ARSCL and KAZRARSCL products rely primarily on lidar-based cloud detections in identifying insect-only events, so we don't expect any systematic differences in the cloud fraction determined by the two products as a result of insects.2) CF from merged ceilometer-MPL uses CBH only and also has the advantage of low missing data as either ceilometer or MPL are used to determine CBH. 3) CF from the ceilometer alone is the most common method that only uses CBH information, but has a disadvantage of missing data and limited vertical range for detecting high level cloud (Table 1). Additionally, total cloud fraction CF is calculated from merged ceilometer-MPL for any cloudy return detected within the narrow-FOV vertical column to screen multiple cloud layers (Sect. 3.4), this merged product has an extended range to 10 km from the MPL.

Narrow-FOV observations of CF, CBH, and wind are computed for two averaging periods: 1) a 30-min period centered on the 15-min averaging time for FSC, meaning the CF time period begins 7.5 minutes before the FSC averaging time-bin and ends 7.5 afterwards; and 2) a 60 min period. The extended observation time (30-min instead of 15-min) for the CF measurement is aimed to compensate for the additional sky area observed by the wide-FOV TSI during a 15-min period.

Hourly CF observations and FSC averages are computed over identical averaging periods for consistency with previous studies.

## 3.2 Wide-FOV Fractional Sky Cover (FSC)

FSCs obtained from TSI observations within the 100° and 160° FOV are available from routine TSI processing. The 100° FOV is used in this analysis because it was previously established to best correspond to nadir observations due to a reduced
impact of cloud sides on estimated cloud cover (Kassianov et al. 2005). Prior to 2005, the TSI used 20° FOV rather than 100°; therefore, we calculate 100° FOV FSC for an additional period (2000-2005) directly from the TSI cloud mask as the fraction of thin and opaque pixels over total number of pixels. In our calculations, we remove a 25° FOV ellipse surrounding the sun location for consistency with the routine TSI processing. Our product compares well with the available TSI product, only 2% of the calculated FSCs exceed the processed FSCs by more than 0.05. The total (opaque + thin) FSC is calculated for each
cloud mask, and then averaged in non-overlapping 15-min and 60-min intervals. Note that 15-min is consistent with the expected decorrelation time of a ShCu field (Kassianov et al. 2005). The TSI does not contain CBH information therefore selection of single-layer ShCu time periods requires the joint use of active sensors.

## 3.3 Narrow-FOV FSC ("CF-like")

A narrow-FOV FSC is calculated from the TSI cloud mask with the sampling, sensitivity and processing of the wide-FOV
FSC. These calculations are aimed to mimic the narrow-FOV lidar-radar measurements ("CF-like"). The narrow-FOV represents a 3x3 pixel region located at a zenith angle of 20° and azimuthal angle of 315° from each TSI cloud mask. There are three main reasons for selection of this region. This region (1) is close to zenith (the zenith region is obstructed by the camera box), (2) is located outside of the sun circle region during the study hours (local daylight 9:00-18:00), and (3) corresponds to the best agreement with the wide-FOV FSC observations (Wagner & Kleiss, 2016). The TSI image resolution
has been increased from 352 x 288 pixels (original resolution) to 480 x 640 pixels (improved resolution) in August 2011. The narrow-FOV (3x3 pixel region) "CF-like" observation has 4.1° (original) and 2.3°(improved) angular resolution. For comparison, the narrow-FOV ARSCL cloud products, including the lidar-radar CF, have much finer angular resolution (about 0.2°). The corresponding spatial resolution of the lidar-radar CF can be estimated by multiplying wind speed at cloud base height by lidar-radar dwell time (about 10 sec). For example, its spatial resolution is about 100 m for 10 m/sec wind speed.

## 3.4 Data Selection and Quality Control Procedures

Data selection and quality control procedures are applied to the averaged and merged dataset. The scatterplots for 30-min CF and 15-min FSC (Fig. B1) illustrate application of the quality control criteria while resulting data completeness is provided in Table B1. These screening criteria are executed on the times with ShCu identified in Sect. 2 and includes the following steps.

1) Periods with clear (averaged FSC < 0.05) or overcast (averaged FSC>0.95) conditions are removed from this analysis for all variables.

2) Periods with multiple cloud layers are screened using merged ceilometer-MPL data where the CF from all clouds is required to be within 0.1 of the ShCu fraction (Berg et al., 2011) (Fig. B1a)

3) Individual 10 s values from the merged ceilometer-MPL product that were flagged as suspect are excluded from the analysis. Incomplete data contributes to errors in the CF calculation (Fig.B1b), and so only periods with 100% available data are used in the analysis.

4) Periods with questionable TSI retrievals are censored using a threshold on allowed thin clouds. The thin cloud classification of the TSI represents an uncertain pixel, which is "not really clear or cloudy" and typically appears as a thin border surrounding the bright and easily visible ShCu clouds. However, camera degradation and normal automated white balance adjustments of the camera can lead to excessive thin cloud as well as erroneous opaque regions. Manual inspection of these time periods results in a QC criterion that discards intervals with thin cloud fraction greater than 0.3 (Fig. B1c).

5) Additional requirement for comparisons using the ceilometer dataset. Individual observations in which the quality-control flag was marked as good are retained and those with faults are censored from the dataset. Averaging periods with missed or censored ceilometer data are discarded. The ten-year (2000-2010) period has a substantial (~40%) fraction of missed ceilometer data (Table B1).

Initial selection criteria for the partially-cloudy events with single-layer ShCu (criteria 1 and 2) reduced the dataset from 614 to 609 days. Quality control (steps 3 and 4) on the 15-min averaged data further reduced it to 569 days, and from 2393 to 2048 hours (Table B1). It is important to note that this dataset is not strictly selective against cases that transitioned from another cloud type (i.e. stratus to cumulus), nor is it selective against cases that are driven by large-scale weather systems.

## 4 Results and Discussion

### 4.1. Cloud detection and data merging

Instrument-dependent values of the estimated CF (Table 1, Sect. 3.1) and the narrow- and wide-FOV FSCs (Sect. 3.2, 3.3) are generated for the whole 18-yr period (2000-2017) and two sub-periods (2000-2010 and 2011-2017). The comparison of the cloud cover estimates from the sub-periods illustrates the joint and individual impacts of instrumental upgrades (Sect. A.2 - A.3) and data merging schemes (Sect. A.4) on the long-term statistics of cloud cover. The comparison includes both 15-min (e.g., Lareau et al., 2018) and 60-min (e.g., Zhang et al., 2017) averaging windows to assess the impact of the shorter averaging time on the CF-FSC agreement.

We begin our investigation of differences in cloud cover estimates by comparing mean values (Tables 2 and 3). The 60-min average FSCs obtained for the two subperiods are comparable (0.34 vs. 0.33), and their distributions are very similar (comparison not shown) indicating weak changes in the mean ShCu cloud cover for the two sub-periods. The ceilometer-based

CF supports this interpretation as mean values show good agreement with FSC, particularly for the later sub-period (2011-2017) where the ceilometer data have good completeness. However, the mean CF calculated from the merged instruments are larger than FSC by 0.12 for the ceilometer-MPL, and 0.08 for the merged lidar-radar in the early sub-period (2000-2010). The larger values of the CF from the merged data in comparison with the ceilometer and FSC in the early sub-period are consistent

with previous studies which focus on all cloud types and sky conditions (Boers et al., 2010; Qian et al., 2012; Wu et al., 2014; Kennedy et al., 2014). However, the FSC-CF correlation obtained for the single-layer ShCu is greatly improved over those from the previous studies with all cloud types. For example, Wu et al. (2014) reported a correlation coefficient of 0.54 comparing hourly CF and FSC when ignoring clear and overcast conditions. Here, the correlation coefficients for hourly data are much higher (0.8-0.84) (Table B3). For the later sub-period (2011-2017), the mean CF from the merged instruments

decreases by 0.05 for the combined ceilometer-MPL and 0.09 for the combined lidar-radar. As a result, the combined lidar-radar CF shows an improved agreement with the FSC, whereas the CF from ceilometer-MPL is still larger than the FSC by 0.08 (Table 2).

The changes in mean values of the merged instrument CFs between the two subperiods are examined in the 1D and 2D histograms in Fig. 1. The figure shows strong correlation between merged ceilometer-MPL and merged lidar-radar CFs in

the early sub-period, indicating that the cloud top height threshold employed has little influence on cloud fraction. In contrast, in the later sub-period the RMSD between the CFs is twice as large and the correlation is only moderate, indicating a higher influence of the cloud top height criteria. The 1D histograms (Fig. 1a,b) show that both variables shifted to have more observations of low CF in the second subperiod, although this is most pronounced for the lidar-radar. These results suggest that changes in merging the ceilometer-MPL cloud base height detection impacted CF moderately, whereas changes in cloud

top height detection in the merged lidar-radar data created significant differences in CF between the two periods.

The 2D histogram comparisons of merged lidar-radar CF and FSC (Fig. 2) show the elimination of bias across the entire range of FSC in the later sub-period; however, this may be due to the introduction of compensating errors when the overprediction of clouds by the combined MPL-Ceilometer cloud base height is compensated by the under prediction of clouds by the radar cloud top height in the later period when only radar is used to detect cloud top height below 3 km. Though the

mean bias is reduced by 0.08 (Table 2) in the later period, the scatter about the 1:1 line is little affected as evidenced by only a 0.04 reduction in the RMSD in the later sub-period; additionally, 51% of CF values differ from FSC by more than 0.1 in the early period compared to 41% in the later sub-period. The modest improvement in agreement indicates the continued presence of cloud detection differences. The 2D histograms for the merged-lidar data CF vs FSC (Fig. 3) show better agreement across the entire range in the later sub-period, consistent with an improvement in ShCu cloud detection. The compensating error

hypothesis is supported further by the comparisons of ceilometer CF to FSC (Fig. 4), which return the highest correlation coefficients (Table B3) and the lowest RMSD (Tables 2,3).

The new cloud radar (deployed in 2010) is expected to have improved ShCu detection in the lower atmosphere over the previous instrument. Therefore, it is not expected that the changes in radar detection are responsible for the *reduction* of merged lidar-radar CF in the later sub-period. We should note that the small droplet size of continental cumuli has been reported

to impede their detection by both the new (Lamer and Kollias, 2015) and retired radar (Chandra et al., 2013). For example, Lamer and Kollias (2015) report that 37% of the hydrometeor detections by the ceilometer were missed by the radar at the SGP site. Most certainly, in either period, the radar would not contribute to an overestimation of CF. Instead it is expected that changes in how MPL and radar data are merged to retrieve cloud top height is likely the source of these significant differences seen between sub-periods (Sect. A.4). Though a number of differences exist, the incorporation of MPL data (below 3 km) in

the original cloud top height retrieval would increase the number of detected cloud tops compared to those retrieved from the radar data alone for the initial period (2000-2010). Reliance only on the radar data for cloud top detection in the updated algorithm would result in fewer cloud top height detections and therefore a lower CF (see Sect. A.4 for more details).

        Since ceilometer CF does not demonstrate dependence on sub-periods it is unlikely that the decrease in merged ceilometer-MPL CF between the two periods is due to the ceilometer. The merged ceilometer-MPL CF is always the same or

greater than CF from the ceilometer alone, and the bias decreases between the two sub-periods from 0.19 to 0.11 (comparing times with complete ceilometer observations). This decrease in bias also corresponds to a decrease in the RMSD from 0.25 to 0.16, and an increase in Pearson's correlation from 0.86 to 0.91. The improvement in agreement might be attributable to improvements in the MPL mask that reduce false cloudy returns from boundary layer aerosols, resulting in a lower MPL CF (Sect. A3). It is also consistent with an increased reliance on the ceilometer for boundary layer retrievals in the cloud base best

estimate retrieval algorithm (Sect. A.4). Specifically, the updates of the merging algorithm would likely result in more frequent "clear" returns. The TSI and ceilometer data in comparison with the merged ceilometer-MPL and/or ceilometer-MPL-radar data may be preferable for consistent estimates of ShCu cloud cover at the SGP site from 2000 to present.

### 4.2. Impact of FOV on cloud cover measurements

        Recall that the CF obtained from lidar-radar observations with narrow FOV represents a transect of a cloudy sky along the

wind direction, while FSC acquired from wide-FOV TSI data defines an area of cloudy sky. Both the CF and the FSC are widespread measurement-based estimates of cloud cover. Fractional sky cover (FSC) from a 100° image spans about a 3 km width of the sky for a 1.5 km cloud base, whereas the active remote sensing instruments (radar, MPL, and ceilometer) observe only a "soda straw" approximately 10-m wide – essentially a point in the sky. Cloud fraction obtained from these narrow field of view instruments is thus essentially a 1-D transect through clouds that pass overhead. Previous studies have reported

sampling uncertainties in transect measurements for modeled random cloud fields (Astin et al., 2001; Berg and Stull, 2002; Kassianov et al., 2005). The term "random" refers to the random arrangement of clouds on a horizontal plane within a given domain. In particular, the previous model studies (Astin et al., 2001; Berg and Stull, 2002) have demonstrated that the cloud cover obtained from the transect measurements mimics the area-averaged cloud cover for non-organized (e.g. random) cloud fields well if the sample size is relatively large (or numerous individual clouds are sampled). Recently Oue et al. (2016) have

showed that 10 or more ceilometers equally spaced across a 25 km width in the cross-wind direction are required to estimate the simulated cloud cover in the small (30 km) domain. Certainly, the number of ceilometers, their locations and averaging time required for an accurate estimation of the cloud cover depend on the spatial arrangement of clouds and wind speed.

Conversely, poor agreement is expected for organized cloud fields, such as so-called "cloud streets," where individual clouds are arranged in rows along the mean wind direction within a given period of interest. To estimate the relative impact of the

FOV configuration on the estimated cloud cover, we use the "CF-like" observations (Sect. 3.3) following an approach previously introduced by Wagner and Kleiss (2016). Recall that the narrow-FOV "CF-like" and the wide-FOV observations are both from the TSI and thus have identical sensitivities for ShCu detection. Thus, the differences between "CF-like" and FSC illustrate the FOV impact on the estimated cloud cover.

Comparison of "CF-like" and FSC shows symmetric variability around the 1:1 line (Fig. 5), and no indication of bias

in average values (Tables 2 & 3). For the 1 h (30 min) observations, 23% (41%) of "CF-like" values differ from FSC by more than 0.1, and 5% (14%) differ by more than 0.2. Reducing the averaging time from 1 hr to 30 min increases the RMSD by 0.05 (Table 2) and decreases the correlation from 0.92 to 0.85 (Table B3). Similar trends in the RMSD, and correlation coefficients are seen for all the CF measurements as well. This result is consistent with a 15-min decorrelation time for the temporal FSC fluctuations (Kassianov et al., 2005). The "CF-like"-FSC comparison allows us to estimate that narrow-FOV observations are

responsible for a ±0.1 uncertainty of for hourly observations of CF (Table 2).

Using the passive "CF-like" observations, we also emphasize the relative impact of the instrument-dependent cloud detection and data merging for actively sensed CF measurements. The variability about the best fits in Figs. 2-4 (comparing FSC to CF) is greater than is expected from FOV impacts alone (Fig. 5) for both hourly and sub-hourly observations. For hourly measurements, 23% of the "CF-like" values differ from FSCs by more than 0.1, this is the expected impact of FOV

alone. For the earlier sub-period (2000-2010), the percentages of the CF values differing by more than 0.1 from the wide-FOV FSC are 62%, 51%, and 34% for the merged ceilometer-MPL, lidar-radar, and ceilometer retrievals, respectively. For the later sub-period (2011-2017), which has the least bias, the corresponding percentages are 53%, 41%, and 30%. The large percentages of observations with significant uncertainty in both time periods highlight the importance of distinguishing between FOV effects from instrumental detection differences when using data for sub-hourly applications.

The effective spatial area sampled by either narrow or wide FOV instruments is a function of both sampling duration and wind speed. High wind speed in comparison with low wind speed (1) increases sample size for a given period and (2) tends to organize horizontal arrangement of clouds (e.g., Weckworth et al. 1999, Atkinson and Zhang 1996). These two factors associated with sample size and spatial arrangement of clouds should be considered when differences between cloud cover obtained from narrow- and wide-FOV observations as function of wind speed are considered (Fig. 6). In particular, Figure 6

illustrates that both CF-FSC and "CF-like"-FSC differences are reduced noticeably as the wind speed increases from 1 m/s to 3 m/s, and continue to reduce slightly as the wind speed grows up to 11 m/s. The CF-FSC and "CF-like"-FSC differences obtained at a higher wind speed (above 11 m/s) should be considered with caution due to limited number of the corresponding cases with high wind speed (e.g., fewer than 100 cases for 60-min time average). The increased sampling area associated with increased wind speed does not necessarily result in an improved agreement between the narrow- and wide-FOV observations

for both hourly and sub-hourly observations due to the impact of wind speed on cloud organization.

### 4.3 Heuristic tool to evaluate individual cloud cover estimates.

We developed a tool to help understand the impact of different sources on individual CF measurements. These sources include data quality control, detection differences, and spatial cloud organization during the time period of interest (FOV-impact). Many factors contribute to cloud field organization including cloud cover, cloud size (Zhu et al., 1992), wind speed, cloud growth and decay rates, and the presence of cloud streets and clusters. The "quick-look" tool is used to distinguish the detection-based uncertainty from the FOV-impact during different sky conditions.

TSI cloud mask images are analyzed for FSC differences along the cross-wind direction to visually assess the spatial inhomogeneity of the cloud field and its potential impact on disagreement between narrow- and wide-FOV observations. Prior to our analysis, the 25° FOV sun circle is removed, the cloud mask is cropped to the 100° FOV and undistorted from hemispherical to rectilinear coordinates (Chow et al., 2011). Mean composite images are generated by summing all images within 15-min and dividing by the number of images (Fig. 7a). Each pixel in the averaged image can be interpreted as a 15-min CF measurement from a narrow-FOV sensor. The variability of CF in the cross-wind direction can indicate the possible influence of cloud field organization on cloud cover estimates by narrow-FOV observations. The composite image is then divided into 21 equally spaced "lanes" parallel to the wind direction at CBH (lane FOV is 5°). The mean and interquartile CF from each lane is then computed for this 15-min average. Figure 7b shows that the cross-wind variability represented by the lane CF means exceeds substantially the within-lane variability (vertical bars). The low variability within each lane indicates small changes of FSC within a lane for this period.

Figure 8 shows an example of "quick-look" results generated for one day (June 14, 2017). The narrow-FOV (5°) lane-FSC (Fig. 7) is used to estimate the FOV impact for each 15-min interval. The variability in lane-FSC for each 15-min interval is visually displayed as one row in the heatmap (Fig. 8a), and provides the spatial distribution of the narrow-FOV FSC perpendicular to the wind direction. The source of between lane-FSC variability can be inferred from the 15-min mean composite images themselves (Fig. 8b). For perfect data, the thumbnail images are a representation of the mean cloud field within the 100° FOV image. For typical windspeeds (4-15 ms-1) it is common to observe streaks, as is seen in this example day, generated by the motion of clouds (see also Fig. 9a, b). For low wind speeds (less than 4 ms-1), a mottled field of clouds emerges with pixel CF dependent upon the rate of cloud formation and decay (Fig. 9c). Less variability in cross-lane FSC is observed for large cloud cover (FSC > 0.7) especially for typical winds (Fig. 9d).

The lane-FSC is compared to actual 30-min CF measurements in Fig. 8c. The intention of this plot is to compare the narrow-FOV CF to the range of possible narrow-FOV FSC observed over different sky regions. By quantifying the uncertainty in cloud cover estimates due to the spatial variations in the mean cloud field, we are better equipped to separate FOV impacts from detection differences. For example, a noticeable cloud street appears from about 12:30-13:15 local time, characterized by a wider spread in the lane-FSC min-max and interquartile range (IQR). The mean FSC is approximately 0.5 during this time and the CF varies between 0.25-0.3, and is nearly equal to the lane-FSC minimum. The corresponding thumbnails in Fig. 8b appear non-uniform, suggesting that the CF measurement is impacted by the narrow FOV configuration. In contrast from

13:15 to 14:15 a more uniform mean cloud field is depicted, and the lane-FSC IQR is within 0.1 of the FSC. During this period, ceilometer CF is also within the lane-IQR. Surprisingly, CF from the merged lidar-radar diverges from the ceilometer value for the first time in the day, and exceeds the lane FSC-maximum significantly. We interpret the larger merged lidar-radar CF to be an *over-estimation* of cloud cover associated with detection differences or data merging issues.

Figure 8 is also useful for quality checks as the "quick-looks" are not censored based on the criteria of Sect. 3.4. For example, on this day we can see little evidence of upper-level clouds by the close agreement of the merged ceilometer-MPL total CF (blue) and ShCu CF (red-dashed). The thumbnails illustrate image artifacts such as incomplete removal of sun glare (17:00-19:00), which would cause FSC to be overestimated. In this same period, the images otherwise show relatively uniform cloud cover, therefore $CF_{ceil}$ might be preferred over FSC. Sun glare is common in the TSI, and a method has been proposed to identify and correct its effects on the FSC (Long, 2010). The wind-direction, indicated by the red arrow, is provided to verify cloud motion with observed wind direction. This is particularly helpful if missing wind data at CBH has been imputed from other sources, as it can be incorrect. Sun glare and incorrect wind direction are significant limitations of this method's extensibility to a statistical analysis.

There are two main expected applications of the introduced "quick-look" tool. The first potential application is a classification of spatial organization of cloud fields using, for example, cross-wind cloud field variability (e.g. peaks and valleys in Fig. 7b) and within-lane variance of cloud amount (e.g. vertical bars in Fig. 7b). Numerous images generated by the "quick-look" tool (e.g., Figure 8b) for the extended period (2000-2017) can be considered as a valuable training dataset for machine learning with focus on automated detection of desired features of the cloud fields (e.g., "cloud streets") and unwanted contaminations of TSI images (e.g., Figure 9). Second potential application is a visual inspection of the generated images for a given period of interest (e.g., a short-term field campaign) to check for the impact of instrumental detection differences and cloud field organization on the observed cloud amount. Visual inspection may be feasible given a limited number (about 40) of ShCu events annually during the warm season. For example, a spread of the lane CFs (gray region in Fig. 8c) gives an idea about the cross-wind cloud field variability within a given FOV, and thus aids in understanding the difference between cloud amounts obtained from the narrow- and wide-FOV observations.

## 5 Conclusions

We compare single-layer ShCu cover estimates obtained from the narrow-FOV lidar-radar and the wide-FOV TSI data collected at the continental SGP site. The data represent an extended period (2000-2017 summers) and two sub-periods (2000-2010 and 2011-2017 summers), which mark the instrumentation upgrades and corresponding improvements in data merging. Our comparison includes the bias between the long-term mean values of the cloud cover estimates, the corresponding RMSDs, and correlation coefficients. The main conclusions are organized along the two guiding questions highlighted in the introduction:

1) *Have significant changes in the observations of ShCu cover occurred at the SGP site due to instrumental and algorithmic upgrades?* We demonstrate that the best agreement (bias and RMSD are within 0.04 and 0.12, respectively) occurs between the CF obtained from the ceilometer alone and FSC obtained from TSI data for the entire period and two sub-periods. In contrast, the CF calculated from combined ceilometer-MPL data has the largest disagreement with FSC in the first sub-period (positive bias of 0.12, RMSD of 0.2), with improvement in the second sub-period (positive bias of 0.08, RMSD of

0.17). This improvement is likely associated with updates to the MPL cloud mask, and improved merging strategies that rely more on the ceilometer. When incorporating a threshold on the cloud top height, the CF obtained from the combined ceilometer-MPL-radar data has better agreement with FSC in the first sub-period (positive bias of 0.08, RMSD of 0.18), and good agreement in the second sub-period (no bias, RMSD of 0.14). The strong period dependence of CF obtained from the combined ceilometer-MPL-radar data, is likely due to a change in what sensors are relied on to detect cloud top below 3 km.

Post 2011, KAZRARSCL stopped using the MPL cloud top detection below 3 km leaving the radar as the sole sensor for cloud detection in that region. However, the radar has known difficulties in detecting cumuli; therefore, the improved agreement is likely due to partial compensating errors in cloud detection.

        2) *What is the impact of FOV configurations on hourly and sub-hourly observations of ShCu cover?* TSI-based narrow- and wide-FOV FSC comparisons have no bias and RMSD of 0.1 for hourly observations for all considered time

periods. The obtained small RMSD shows that, on average, the narrow-FOV introduces an uncertainty in cloud cover of ±0.1 for 1-h CF measurements. The "penalty" for decreasing the averaging time to 30-min (15-min FSC) is to increase the average uncertainty in CF to ±0.15. The majority (77%) of the comparisons are within 0.1 for hourly observations, whereas only 59% of the 30-min (narrow-FOV) versus 15-min (wide-FOV) comparisons are within 0.1. This considerable impact of the FOV configuration on sub-hourly observations of ShCu, confounded with instrumental detection differences motivates the

introduction of a new "quick-look" tool.

        The "quick-look" tool uses a spatially resolved analysis of the cross-wind variability in cloud cover in the TSI 100° FOV to provide an expected range of cloud cover that would be detected by a set of 21-narrow-FOV instruments with the same detection properties as the TSI. We demonstrate the utility of the "quick-look" tool to identify the impacts of the FOV configuration on the cloud cover estimates and to identify periods with organized cloud fields. In addition, the "quick-look"

tool can identify times with potential issues associated with data merging and cloud detection, as well as periods with TSI data quality concerns. The developed data-centric tool may supplement outputs from the existing and future narrow-FOV lidar-radar simulators (Bouniol et al., 2010; Tatarevic and Kollias, 2015; Lamer et al., 2018).

        The presented dataset provides a unique opportunity to (1) contrast the long-term cloud cover estimates with different instrumental sensitivities and data merging strategies and (2) describe the across-wind variability of cloud cover at user-

specified temporal and spatial scales. These data provide an observational foundation for a better interpretation of quandaries such as model-to-data discrepancy of cloud cover (Zhang et al., 2017). Finally, the developed dataset is expected to facilitate evaluation of the modeled shallow cumuli over small domains (~4 km x 4 km) at the SGP site. These studies with strong model and observational components include the recent large eddy simulation (LES) ARM Symbiotic Simulation and Observation

(LASSO) project (Gustafson et al., 2015) and the Holistic Interactions of Shallow Clouds, Aerosols, and Land-Ecosystems
(HI-SCALE) field campaign (Fast et al., 2018).

**Data Availability**

The analysis dataset is derived from observations collected at the Atmospheric Radiation Measurement (ARM) Southern Great Plains (SGP) site Central Facility in northern Oklahoma, USA and are freely available at https://www.arm.gov/data. Specific data streams in Sect 2 can be requested through the interactive search tool, or accessed through the DOI in the data reference.

The data from the figures, and novel quick look tool have been made available for all 614 days with detected shallow cumuli on the ARM archive (https://www.archive.arm.gov/) as principal investigator data (PI reference name is Kleiss): TSI composite images merged cloud fraction product for shallow cumulus cases (tsiQLtable). Searchable by DOI: 10.5439/1523254.

**Appendix A: Description of Millimeter Cloud Radar (MMCR), Micropulse Lidar (MPL), and merging algorithm 2000-2017**

**A.1 Active Remotely Sensed Cloud Locations product Overview**

MMCR, MPL, and ceilometer data are combined in the Active Remotely-Sensed Cloud Locations (ARSCL) value added product (Clothiaux et al., 2000). The product improves upon the MMCR vertical profiling of hydrometeors by employing an MPL cloud mask to identify CBH and aide in mitigating the significant impact of insect returns. The screened high temporal (10 s) and vertical (45 m) radar reflectivity best estimate (Kollias et al. 2016) is further distilled to generate a user-friendly data stream of the vertical distribution of hydrometeors above the SGP site (sgparsclbnd1clothC1.c1). Within this data stream is a field for best estimate of CBH ("CloudBaseBestEstimate") from merging MPL and ceilometer data, and two fields containing up to 10 hydrometeor layer determinations of cloud bottom ("CloudLayerBottomHeightMplCloth") and top ("CloudLayerTopHeightMplCloth") heights. These two fields are derived from the reflectivity best estimate, cloud base best estimate, and MPL cloud mask. In 2011, the ARSCL was replaced with the KAZRARSCL with corresponding data stream name "sgparsclkazrbnd1kolliasC1.c1". Although the reporting interval of the MPL and ceilometer are 30 s and 16 s respectively, the ARSCL reporting interval is 10 s corresponding to the MMCR, and 4 s for KAZRARSCL corresponding to the KAZR.

Each instrument (MMCR, MPL and ceilometer) has challenges for detecting ShCu, therefore the data merging process can significantly impact the resulting cloud fraction. Radar reflectivity is weak for shallow cumuli with low liquid water path (below $50 gm_{-2}$) due to the small cloud droplet radii (3-7 μm); for this reason, approximately half of the clouds identified by the ceilometer are not seen by the radar (Chandra et al. 2013). Consequently, the lidar data (ceilometer and MPL) are used for CBH information.

The MPL has a greater sensitivity to cloud droplets than the ceilometer and MMCR, resulting in a larger CF (Donovan and van Lammeren, 2001; Kennedy et al. 2014). Some weaknesses of the MPL include 1) misidentification of aerosol layers as clouds, due to the increased sensitivity to aerosols of shorter wavelength of the laser, 2) difficulty retrieving cloud bases in the "blind spot" below 500 m (Welton and Campbell, 2002; Sivaraman and Comstock, 2011), 3) quick attenuation of the backscatter signal within clouds resulting in inaccurate cloud top height, 4) Longer integration time (30 s) may complicate returns in partially cloudy conditions. It was previously found that when MPL is merged with the ceilometer, the combined lidar CF is larger by 0.03-0.06 than CF from the MPL alone (Kennedy et al. 2014). The ceilometer has better visibility and accuracy in the lower atmosphere than the MPL, but its range is limited to 7.7 km. Below 3km, the ceilometer is used to define CBH if both MPL and ceilometer detect clouds, otherwise if just one instrument detects cloud then a CBH is returned from that instrument. Therefore, there is a preference to report a 10s period as cloudy if the lidars disagree.

The lidars are limited in determining cloud top height owing to the attenuation of the backscatter signal. Cloud top height from the MPL is typically assessed by setting a threshold on the attenuation of the return; for this reason, MPL cloud top heights are usually considered effective top heights. Multiple cloud layer detection is better performed by the radar and the algorithmic updates in KAZRARSCL processing reflect this strength (Sect. A4).

**A.2 Radar Updates (2011)**

Many continental cumuli are optically thin (liquid water content less than 50 g/m2) and have small droplets. As a result, the cloud radars (both MMCR and KAZR) are not able to detect a fraction of these clouds (e.g., Chandra et al., 2013; Lamer and Kollias, 2015). The ARM radars have undergone a number of upgrades over the years as was recently reviewed by Kollias et al. (2016) (see their Table 17-2). The millimeter cloud radar (MMCR, 1997-2010) was replaced by Ka-Band Zenith Radar (KAZR, 2011-present) with expected improved sensitivity to cloud properties. The KAZR differs from the MMCR particularly due to the change in radar pulse compression type from Barker to non-linear frequency modulation modes, which improve the radar's sensitivity. Importantly, the cirrus mode (pulse length of 8000 ns) was dropped from the pulse sequence and a more sensitive moderate mode (4000 ns) was added. The MMCR and KAZR both employ a general mode (300 ns). The shorter pulse widths serve to increase the radar's visibility in the lower atmosphere. This is a consequence of an expanded range in the lower atmosphere that is gained when the radar's transmission time is decreased. Another key feature of the upgrade was a transition from sequential pulse sequences employed in the MMCR to dual acquisition pulses in the KAZR. The sequential pulse sequence required a sampling period of 36 s in the earliest period prior to 2004, then 12-14s in 2004-2010 to cycle through the different modes. The KAZR only employs two modes (general and moderate), which can be transmitted in tandem, this reduces the sampling period to 4 s. The improvement in sampling period is reflected in the shortened reporting interval in the KAZRARSCL data product. Cloud radars (both the MMCR and KAZR) are known to miss detection of ShCu with small droplets compared to the ceilometer. For example, Chandra et al (2013) have found that the MMCR misses the majority of continental ShCu observed at the SGP site.

## A.3 MPL Updates (2010)

The MPL cloud mask calculates cloud boundaries predominately from the attenuated backscatter signal. The original cloud mask involved a signal-to-noise threshold based on cloud droplet scattering (Campbell et al. 1998), whereas the updated cloud mask incorporated information on different scattering properties of aerosol and clouds (Wang and Sussen 2001). In 2004 the MPL non-polarized implementation was replaced with a polarized system. The MPL hardware was upgraded in 2010 to a fast switching mode that allowed for switching between linear and circular polarization channels. The cloud mask algorithm was also updated at this time to a methodology developed by Wang and Sassen (2001) which has been used to process the entire MPL record at the ARM facilities, though this cloud boundary processing has only been used in ARSCL and KAZRARSCL since 2010. It is expected that the updated cloud mask would improve cloud returns. Interested readers are directed to the technical documentation for the MPL mask (Sivaraman and Comstock, 2011).

## A.4 Cloud Detection Algorithm Developments (2011)

Many changes occurred between the summers of 2010 and 2011: 1) ARSCL was changed to use the MPL cloud mask of Wang and Sassen (2001) in 2010, 2) the new KAZR was deployed, and the MMCR was decommissioned, and 3) KAZRARSCL replaced ARSCL in 2011 including updates to how retrievals of cloud layer boundaries (bases and tops) are handled in the MPL cloud mask. It is beyond the scope of this paper to identify all the possible changes that could result in differences in the ARSCL- and KAZARSCL-based cloud detections. However, it is important to note that both the MPL and MMCR are used to determine the cloud top heights within the boundary layer in ARSCL, whereas in KAZRARSCL only the KAZR is used to determine cloud top heights below 3 km due to well-known MPL difficulties detecting low clouds (Berg and Kassianov, 2008; K. Johnson, private communication, 2018). In KAZRARSCL the MPL mask is still used for removing clutter in the radar returns.

Updates to the cloud base best estimate algorithm are also implemented. In particular, when the ceilometer observed clear skies and the MPL had no return, the ARSCL data is flagged as "possibly clear" whereas the KAZRARSCL data are flagged as clear. The ceilometer is used to judge clear or cloudy below 500 m in KAZRARSCL. Otherwise in both versions the ceilometer's cloud base is preferred if both instruments return cloud below 3 km, and if either detects cloud then a cloud base height is returned (with the exception noted above).

## Appendix B. Supporting Figures and Tables

Scatter plots of sequential application of quality control procedures (Sect 3.4) for 30-min CF and 15-min FSC are shown in Fig. B1, while data completeness for each summer in this study is provided in Table B1. The regression coefficients and correlation coefficients for Figures 3-6 are provided in Tables B2 and B3, respectively.

## Author Contributions

All authors contributed to the experimental design, analysis plan, and editing of the manuscript. ER conducted the statistical data analysis. JK implemented the "quick-look" tool. LR and LB advised on the active remote sensing instrumentation. CL advised on the TSI quality control. EK provided project oversight and direction. EK and ER lead the composition of the manuscript.

## Competing Interests

The authors declare that they have no conflict of interest.

## Acknowledgments

We would like to thank radar experts Joseph Hardin and Nitin Bhawadraj at Pacific Northwest National Labs for researching (and many conversations explaining) the updates to the MMCR and replacement by KAZR. We'd also like to thank Karen Johnson for diagraming the changes made to the cloud base best estimate algorithm in the ARSCL processing. We also appreciate all of the technicians, software engineers, data managers, and scientists who work together to produce ARM data, in particular Victor Morris who is in charge of the ceilometer and TSI measurements for ARM that were pivotal to this work. This research was supported by the U.S. Department of Energy's Atmospheric System Research Grants DE-SC0016084 and KP1701000/57131. The Pacific Northwest National Laboratory is operated by Battelle Memorial Institute under contract DE-AC06-76RLO 1830.

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

**Figures**

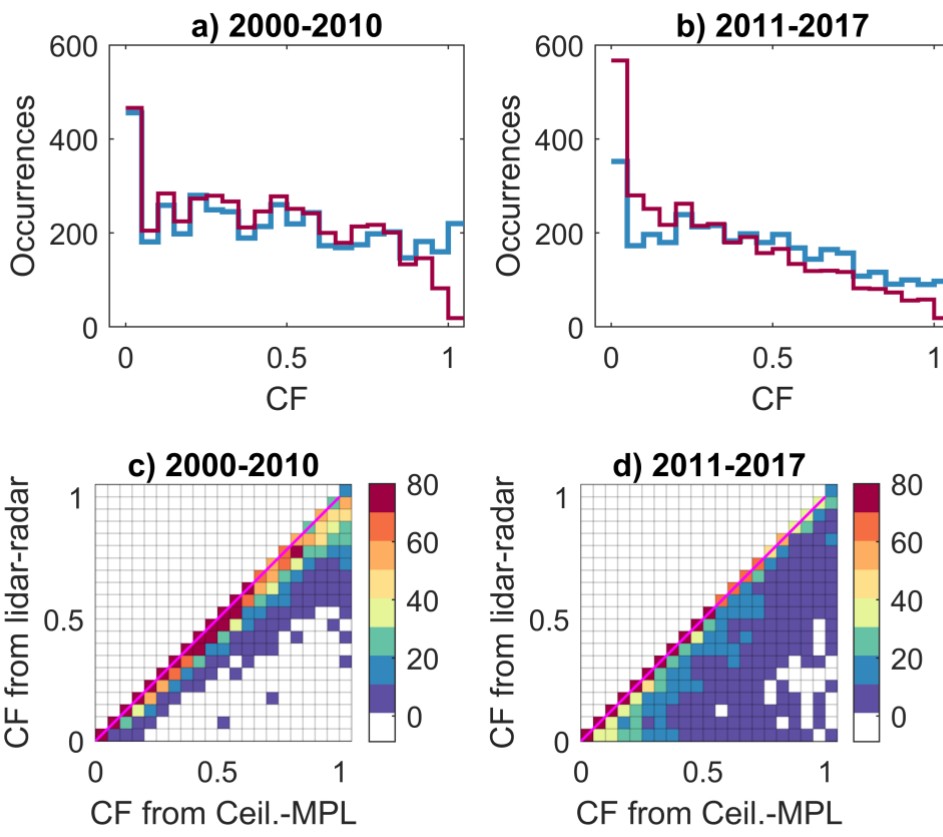

**Figure 1. Relationship between CF from merged ceilometer-MPL (clouds identified by CBH) and from merged lidar-radar (clouds identified by both CBH and cloud top height). (a,b) histograms of CF from merged ceilometer-MPL (blue) and merged lidar-radar (red). (c,d) joint histograms (counts) for the two methods, magenta line is 1:1. (a,c) present data obtained from 2000-2010 (N = 4628), where the radar and MPL data are used jointly to determine cloud top height (Table 1). (b,d) present data obtained from 2011-2017, (N = 3564), where the cloud top height is determined from radar data alone below 3 km. Each observation represents a 30-min average. RMSD (Pearson's correlation coefficient) in (c) is 0.08 (0.98) and 0.17 (0.87) in (d).**

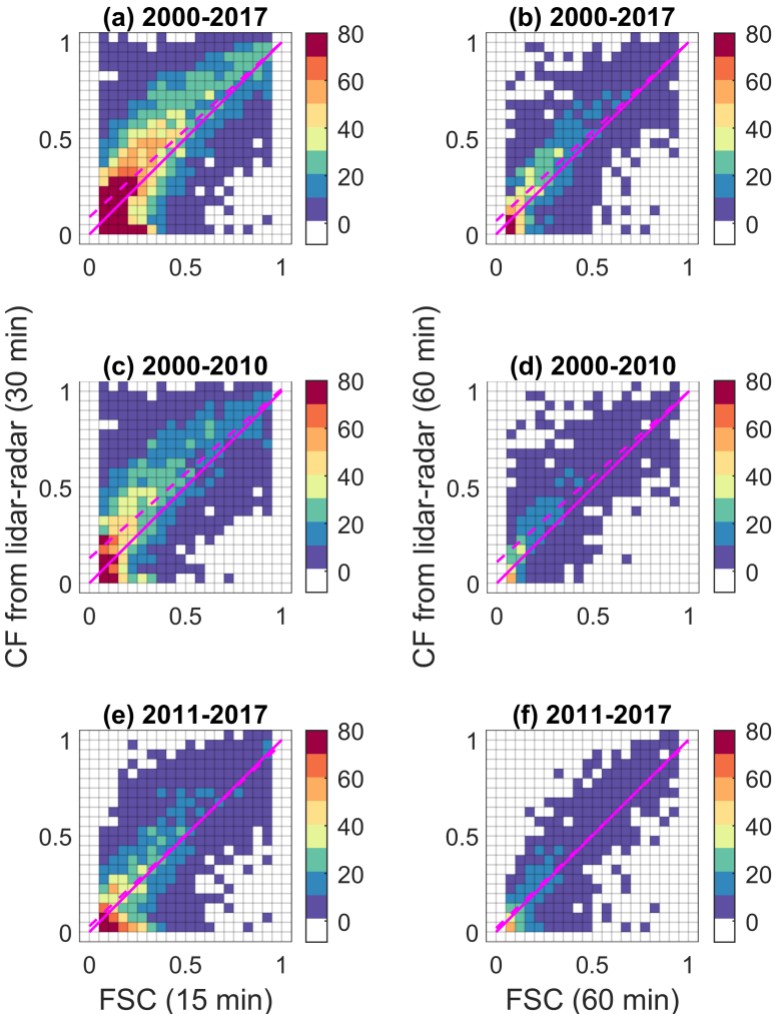

**Figure 2. Two-dimensional (2D) occurrence distributions of paired-in-time CF and FSC for the three periods of interest: 2000-**
**2017(a,b), 2000-2010 (c,d), and 2011-2017(e,f). Solid magenta line is 1:1 and dashed is least squares fit, with regression coefficients given in Appendix B (Tables B2,B3). Left (a,c,e) and right (b,d,f) columns define fine (15-min FSC and 30-min CF) and coarser (60-min FSC and CF) temporal scales respectively. Color scale represents counts in increments of 10. Each included data point excludes clear and overcast conditions using criteria: 0.05 < FSC <0.95. CF from merged lidar-radar method uses information about cloud base height (ceilometer-MPL) and cloud top height (radar-MPL).**

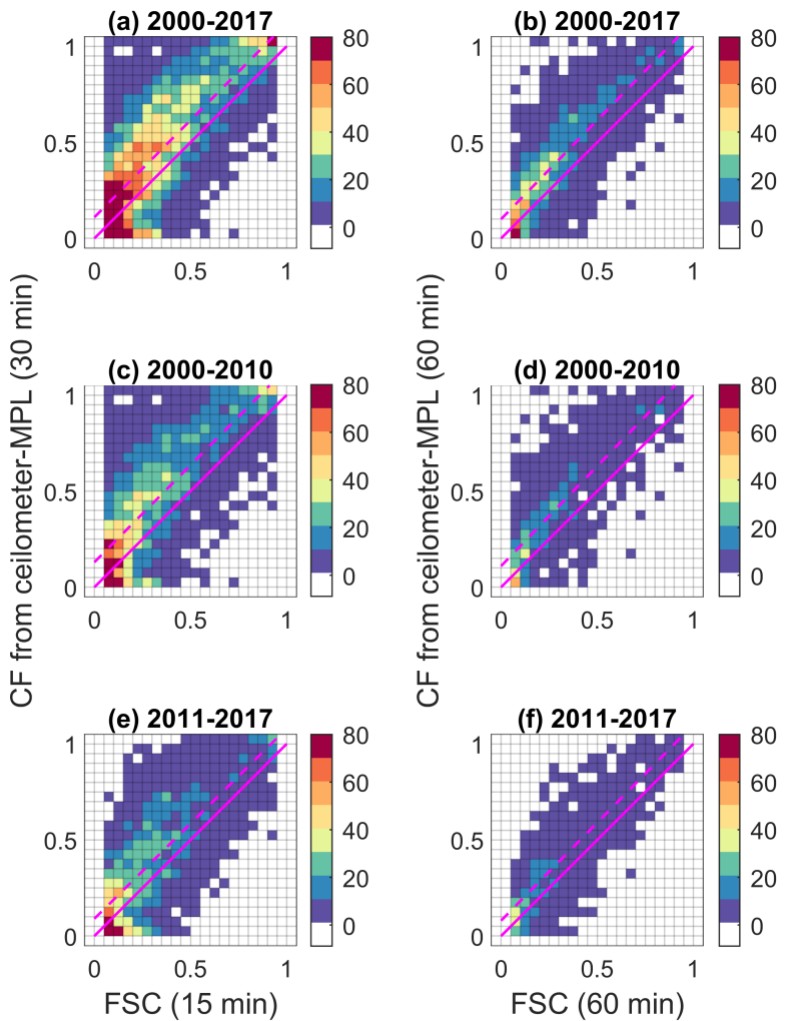


**Figure 3. The same as Fig. 2, except CF from merged ceilometer-MPL method only uses information about cloud base height.**

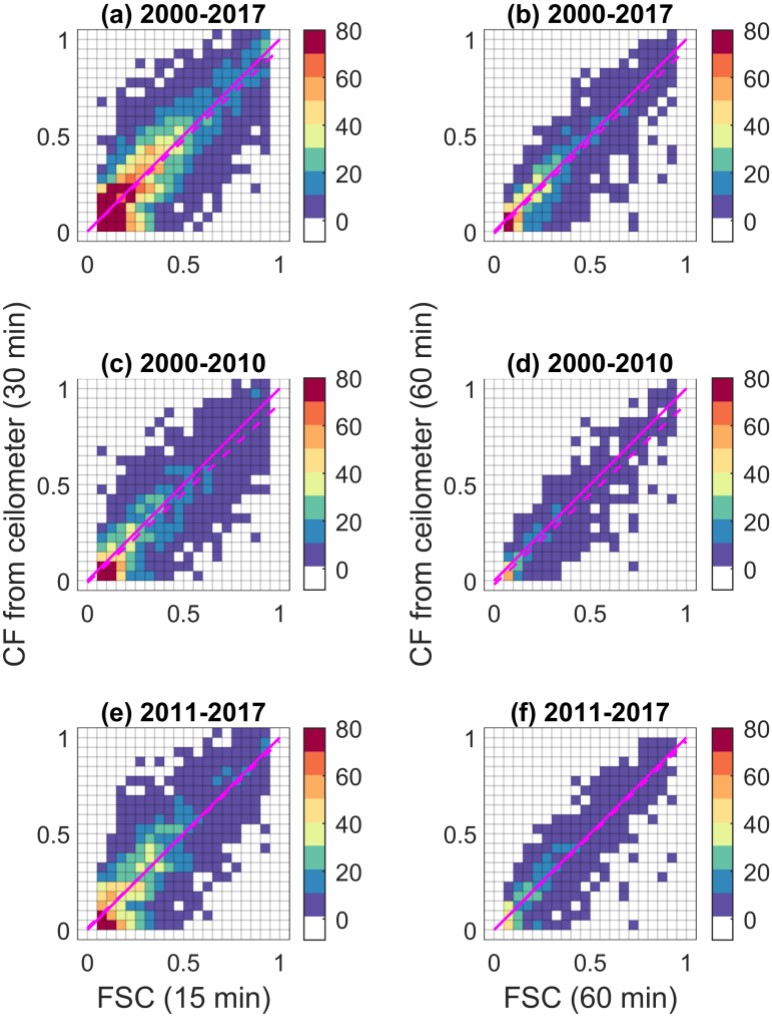

**Figure 4. The same as Fig. 2, except CF only uses cloud base height information from the ceilometer.**

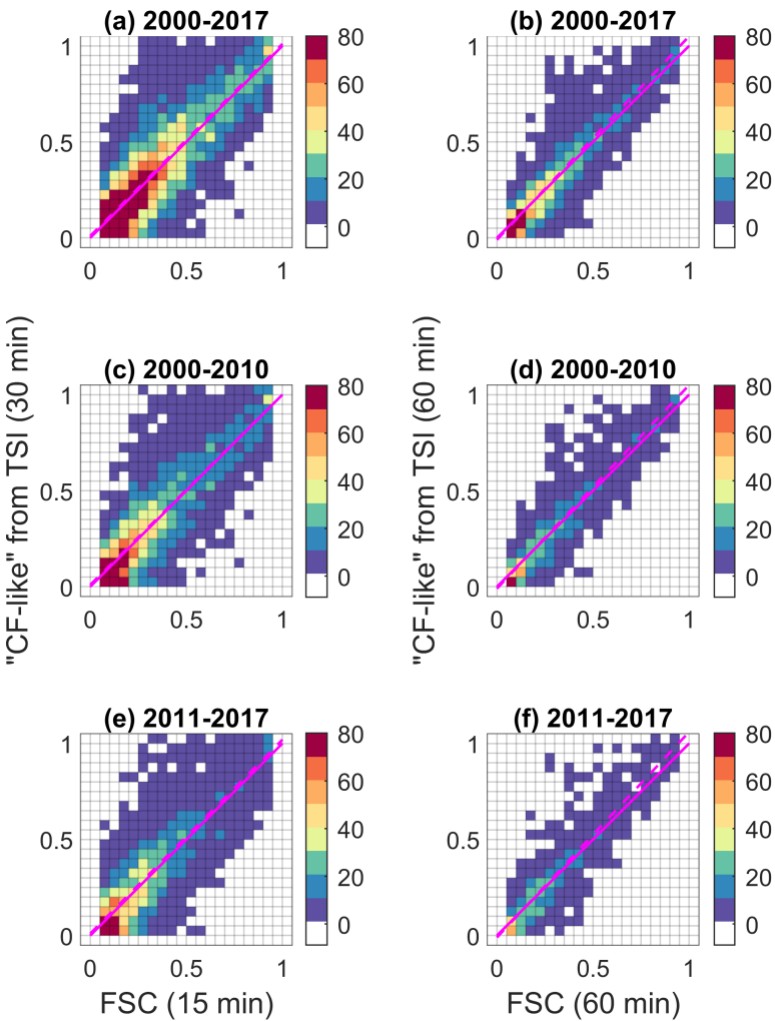

**Figure 5. The same as Fig. 2, except for comparison between "CF-like" and FSC, where both are derived from the TSI instrument.**

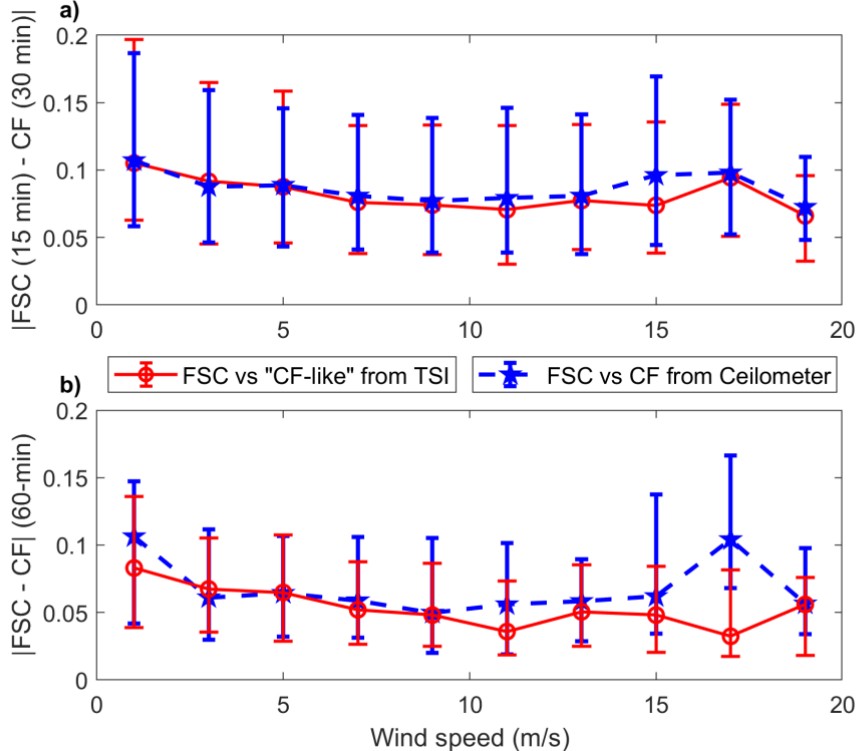

**Figure 6. Absolute difference between FSC and CF as a function of wind speed at CBH (2 m/s bins). Markers indicate median, and bars indicate 25th and 75th quartiles. Red: absolute difference between FSC and "CF-like" from the TSI. Blue: absolute difference between FSC and CF from the ceilometer. a) 15-min FSC and 30-min CF and b) 60 min FSC and CF. Missing wind speed at CBH is replaced by RWP wind speeds at 500 m, missing RWP data is replaced using surface wind speeds.**

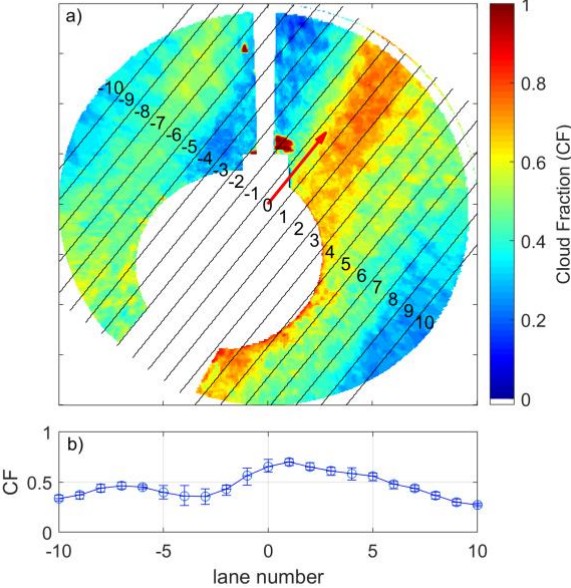


**Figure 7. a) A composite 15-min average cloud cover image (Sect. 4.3) created from 100° FOV TSI cloud masks captured every 30 s with sun circle region removed and projected to a rectilinear grid. Color bar indicates the fraction of time each pixel observes a cloud (thin or opaque). North is top of figure; streaks result from cloud motion across the image field. Image is divided into 21 lanes**
**oriented parallel to the wind direction (red arrow points down-wind) obtained from a radar wind profiler at CBH. b) The mean and interquartile (vertical bars) of CF observed within each of the 21 lanes. Note, the moderate (~20°) misalignment of the cloud streaks and the wind vector, illustrating the challenges of wind-based analyses.**

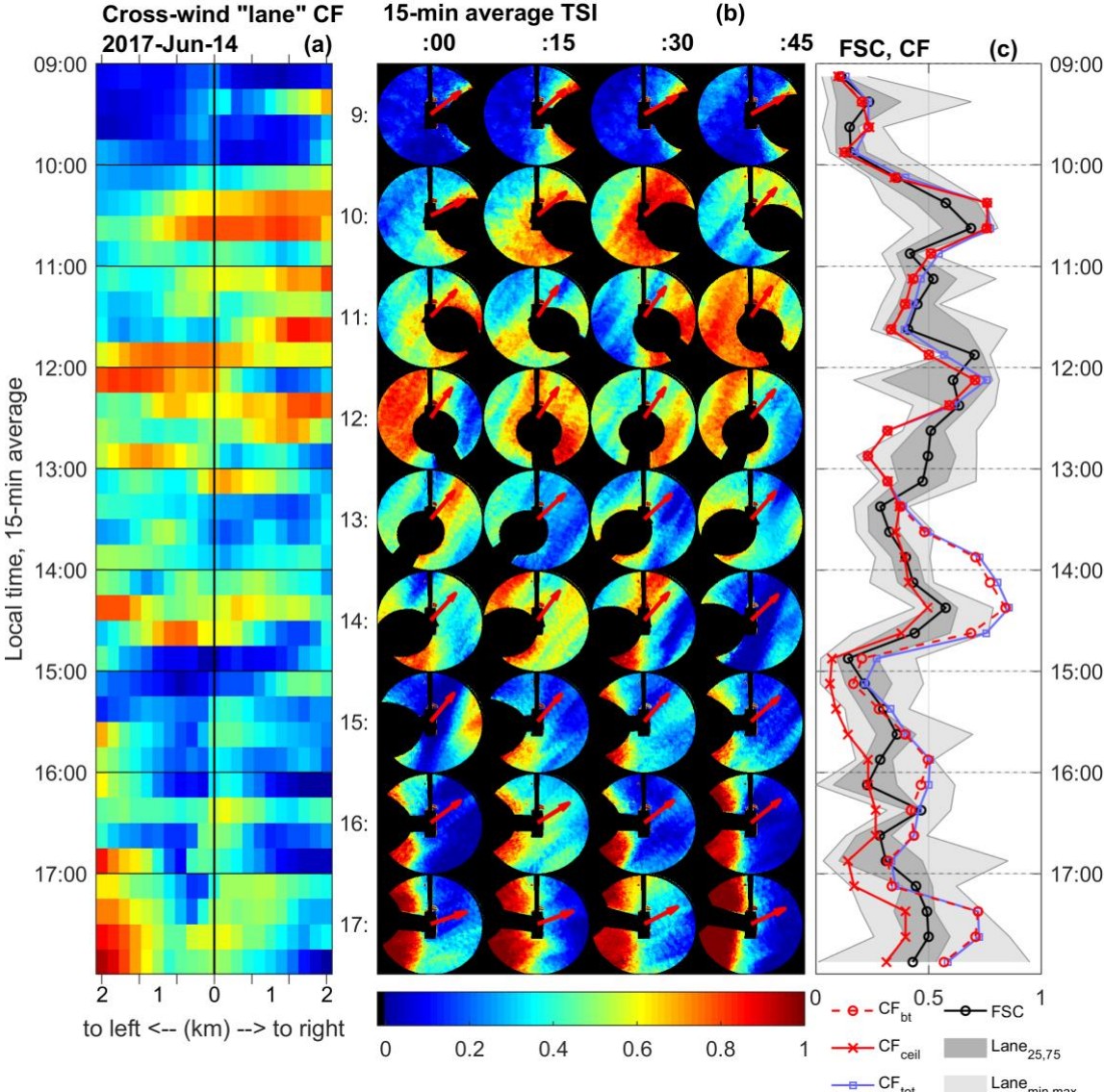

**735**

**Figure 8. Example of results from "quick look" tool for comparing FSC and CF for 14 Jun, 2017. (a) Heat map of 15-min average CF indicated by the color bar. Cross-wind distance is calculated from the average CBH of 1.5 km, with zero representing the centre lane of the composite image (the row from 13:00-13:15 (LST) corresponds to the FSC values in Fig. 7b. (b) 15-min composite images within the 100˚ FOV, projected onto rectilinear grid. Red arrows indicate wind direction at CBH. (c) Comparison of cloud amounts**
**740** **obtained from TSI and lidar-radar data: 30-min averages of CF$_{bt}$ (CF "cloud bases and tops") from merged lidar-radar for ShCu only (red dashed with circles), CF$_{ceil}$ (CF "from ceilometer", red line with "x" symbol), and CF$_{tot}$ (CF "total") from merged ceilometer-MPL for any cloud detection (blue line with squares). 15-min average FSC from TSI data (black with circles), 15-min minimum and maximum lane-averaged CF (light grey shading) and interquartile lane CF (dark grey shading). Markers are placed at time bin centre.**

**745**

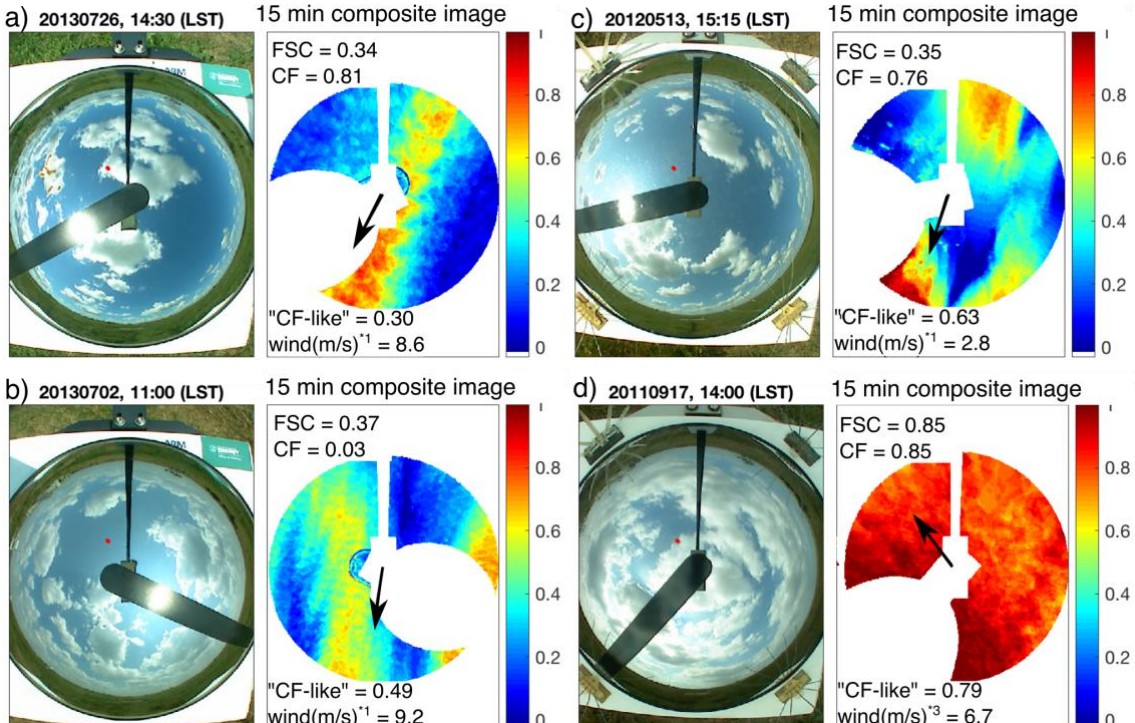

**Figure 9. Illustrations of spatial patterns of ShCu manifest in 15-min averaged composite images (100° FOV, rectilinear coordinates) and a representative all sky image from the averaging period. Color bar is 15-min mean FSC for each pixel in the composite image, black arrow indicates wind direction, red dot in sky image indicates location of the 3x3 pixel for "CF-like". 15-min FSC, 30-min CF from merged lidar-radar method, "CF-like", and wind speed are provided as text. a) A bit of bird feces contaminates the mirror on the left-hand side of the image, fortunately not within the 100-degree FOV resulting in no error of FSC. a-c) *1: wind source is radar wind profiler at CBH, d) *3 is 10 m surface wind speed.**

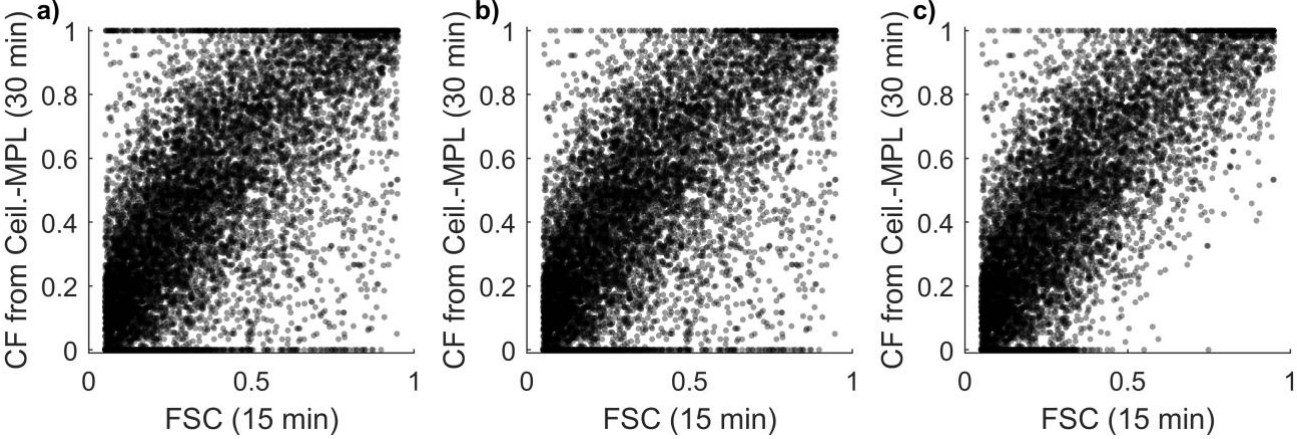

**Figure B1. Comparison of 15-min FSC and 30-min CF from the merged ceilometer-MPL method after data screening steps: (a) Single-layer cases: CF (all clouds) – CF (ShCu) < 0.1 and 0.05<FSC < 0.95 (N = 9571), (b) data as in (a), with additional requirement**

to discard periods with missing CBH returns (N = 9128), (c) data as in (b), with periods when 15-min FSC from thin cloud exceeds 0.3 are removed (N = 8192). Each point represents 15-min period.

**Tables**

**Table 1. Variants of cloud fraction (CF) products used in this work: contributing instruments, thresholds on cloud base height (CBH) and cloud top height (CTH).**

| Instruments used for CF | Abbreviation | CBH thresholds | CTH threshold |
|---|---|---|---|
| Ceil. + MPL† + radar | $CF_{bt}$ (bases & tops) | 0.3 km < CBH < 3 km | CTH < 4 km & CTH > CBH |
| Ceilometer + MPL | $CF_b$ (bases only) | 0.3 km < CBH < 3 km | - |
| Ceilometer + MPL | $CF_{tot}$ (all clouds) | CBH > 0 | - |
| Ceilometer | $CF_{ceil}$ (ceil. only) | 0.3 m < CBH < 3 km | - |

**†In 2011-2017 period, the MPL is not used to determine cloud top height below 3 km (Sect. A4).**

**Table 2. Total number of observations (N) and summary statistics for cloud amount estimated by five methods for three periods of interest (second-fourth rows) at coarser temporal scale: 60-min FSC, "CF-like" and CF. These statistics include the mean (second column), and root mean-square difference (RMSD) (third column). Values more than 0.05 different from FSC are bold.**

| | | | Mean cloud cover | | | | | RMSD: FSC vs CF | | | |
|---|---|---|---|---|---|---|---|---|---|---|---|
| Year | N | $N_{ceil}$ | FSC | $CF_b$ | $CF_{bt}$ | $CF_{ceil}$ | CF-like | $CF_b$ | $CF_{bt}$ | $CF_{ceil}$ | CF-like |
| All | 2250 | 1665 | 0.33 | **0.44** | **0.38** | 0.31 | 0.34 | 0.19 | 0.16 | 0.11 | 0.1 |
| 2000-2010 | 1259 | 708 | 0.34 | **0.46** | **0.42** | 0.3* | 0.34 | 0.2 | 0.18 | 0.11 | 0.1 |
| 2011-2017 | 991 | 957 | 0.33 | **0.41** | 0.33 | 0.32 | 0.33 | 0.17 | 0.14 | 0.12 | 0.1 |

**Each 60 min period excludes clear and overcast conditions using criteria: 0.05 < FSC <0.95. *Note that the ceilometer data in 2000-2010 had a large percentage of missing during quality control (see Table B1), and that the mean is calculated from a subset of times reflected by $N_{ceil}$. CF abbreviations defined in Table 1.**

**Table 3. The same as Table 2, except for fine temporal scale: 15-min FSC and 30-min "CF-like" and CF.**

| | | | Mean cloud cover | | | | | RMSD: FSC vs CF | | | |
|---|---|---|---|---|---|---|---|---|---|---|---|
| Year | N | $N_{ceil}$ | FSC | $CF_b$ | $CF_{bt}$ | $CF_{ceil}$ | CF-like | $CF_b$ | $CF_{bt}$ | $CF_{ceil}$ | CF-like |
| All | 8192 | 6070 | 0.34 | **0.45** | **0.39** | 0.32 | 0.35 | 0.22 | 0.2 | 0.14 | 0.15 |
| 2000-2010 | 4628 | 2586 | 0.34 | **0.47** | **0.43** | 0.3* | 0.35 | 0.24 | 0.21 | 0.14 | 0.15 |
| 2011-2017 | 3564 | 3484 | 0.33 | **0.43** | 0.35 | 0.33 | 0.34 | 0.21 | 0.18 | 0.15 | 0.14 |

**Each 15 min period excludes clear and overcast conditions using criteria: 0.05 < FSC <0.95. *This value is calculated from a much smaller subset of times, $N_{ceil}$. CF abbreviations defined in Table 1.**


**Table A1. References to instrumentation descriptions and service records at the SGP site:**

| Instrument (abbreviation) | Reference |
|---|---|
| Millimeter cloud radar (MMCR, 2000-2010) | https://www.arm.gov/capabilities/instruments/mmcr |
| Ka-band zenith pointing radar (KAZR, 2011-present) | https://www.arm.gov/capabilities/instruments/kazr |
| Micropulse lidar (MPL) | https://www.arm.gov/capabilities/instruments/mpl |
| Ceilometer (Ceil) | https://www.arm.gov/capabilities/instruments/ceil |
| 915 MHz Radar Wind Profiler (RWP) | https://www.arm.gov/capabilities/instruments/rwp |
| Total Sky Imager (TSI) | https://www.arm.gov/capabilities/instruments/tsi |

**Table B1. Summary of single-layer ShCu events by year and percent data completeness after applying quality control steps. Entries**
**correspond to scatterplots in Figure B1. In particular these periods comply to the requirement that 0.05 <FSC < 0.95.**

| Year 20xx | '00$_*$ | '01 | '02 | '03 | '04 | '05$_\dagger$ | '06 | '07 | '08 | '09 | '10$_¥$ | '11 | '12 | '13 | '14 | '15 | '16 | '17 | All |
|---|---|---|---|---|---|---|---|---|---|---|---|---|---|---|---|---|---|---|---|
| Days$_{a)}$ | 12 | 29 | 43 | 28 | 44 | 22 | 36 | 37 | 56 | 43 | 6 | 21 | 33 | 47 | 45 | 30 | 42 | 35 | 609 |
| Hours$_{a)}$ | 52 | 93 | 205 | 124 | 191 | 88 | 135 | 150 | 227 | 155 | 20 | 86 | 98 | 179 | 182 | 112 | 149 | 148 | 2393 |
| QC pval (%)$_{b)}$ | 100 | 70 | 100 | 91 | 93 | 98 | 97 | 96 | 98 | 86 | 14 | 100 | 100 | 100 | 100 | 99 | 99 | 100 | 95 |
| QC thin (%)$_{c)}$ | 91 | 98 | 68 | 88 | 100 | 41 | 100 | 98 | 91 | 86 | 97 | 90 | 96 | 99 | 95 | 79 | 91 | 98 | 90 |
| QC "ceil ok" (%) | 91 | 2 | 0 | 0 | 1 | 1 | 99 | 97 | 91 | 86 | 95 | 90 | 94 | 99 | 95 | 76 | 84 | 96 | 65 |
| Final Days$_{c)}$ | 12 | 20 | 36 | 28 | 41 | 12 | 36 | 36 | 56 | 38 | 3 | 21 | 33 | 47 | 45 | 29 | 42 | 34 | 569 |
| Final Hrs$_{c)}$ | 47 | 63 | 139 | 98 | 176 | 35 | 131 | 141 | 203 | 121 | 2 | 78 | 95 | 178 | 173 | 88 | 135 | 146 | 2048 |

**a) Data in Fig. B1a. b) Percent of observations that pass a 100% Quality Control (QC) percent valid test for ARSCL / KAZRARSCL; data shown in Fig. B1b. c) Data shown in Fig. B1c. *TSI data begins July 2. †TSI missing most data in June and July. ¥In 2010, 19 days with shallow cumuli are identified, of which 6 contained events longer than 2 hours that also satisfied single layer conditions: CF(all clouds) – CF (ShCu) < 0.1). Only 14% of the observations satisfied the 100% instrument validity criterion. The values in this table**
**are generated from the 15-min dataset. These values do not correspond to the 1-hr values (i.e. the 1-hr values are not generated from the 15-min values, and are subject to passing the same quality control thresholds, but on the hourly timescale).**

**Table B2. Regression coefficients for Figures 3-6.**

| | 15- min: $CF = \beta_0 + \beta_1 * FSC$ | | | | | | | | 1 hr: $CF = \beta_0 + \beta_1 * FSC$ | | | | | | | |
|---|---|---|---|---|---|---|---|---|---|---|---|---|---|---|---|---|
| **Year** | **CF-like** | | **CF$_{ceil}$** | | **CF$_b$** | | **CF$_{bt}$** | | **CF-like** | | **CF$_{ceil}$** | | **CF$_b$** | | **CF$_{bt}$** | |
| | ($\beta_0$ |$\beta_1$) | ($\beta_0$ |$\beta_1$) | ($\beta_0$ |$\beta_1$) | ($\beta_0$ |$\beta_1$) | ($\beta_0$ |$\beta_1$) | ($\beta_0$ |$\beta_1$) | ($\beta_0$ |$\beta_1$) | ($\beta_0$ |$\beta_1$) |
| All | 0.01 | 0.99 | 0.01 | 0.92 | 0.12 | 1 | 0.09 | 0.9 | -0.01 | 1.06 | -0.01 | 0.96 | 0.1 | 1.03 | 0.07 | 0.93 |
| 2000-2010 | 0.01 | 0.99 | 0 | 0.91 | 0.13 | 1.01 | 0.13 | 0.87 | -0.01 | 1.06 | -0.02 | 0.94 | 0.11 | 1.04 | 0.11 | 0.89 |
| 2011-2017 | 0.01 | 0.99 | 0.02 | 0.93 | 0.11 | 0.98 | 0.04 | 0.92 | -0.01 | 1.07 | 0 | 0.98 | 0.08 | 1.02 | 0.02 | 0.97 |

CF abbreviations defined in Table 1.

**Table B3. Pearson's Correlation coefficient for FSC vs CF for fine and hourly time scale.**

| Year | 30 min CF, 15-min FSC | | | | hourly CF and FSC | | | |
|---|---|---|---|---|---|---|---|---|
| | CF$_b$ | CF$_{bt}$ | CF$_{ceil}$ | CF-like | CF$_b$ | CF$_{bt}$ | CF$_{ceil}$ | CF-like |
| All | 0.77 | 0.74 | 0.83 | 0.85 | 0.84 | 0.81 | 0.89 | 0.92 |
| 2000-2010 | 0.78 | 0.74 | 0.86* | 0.85 | 0.84 | 0.8 | 0.9* | 0.92 |
| 2011-2017 | 0.77 | 0.77 | 0.82 | 0.84 | 0.84 | 0.83 | 0.88 | 0.92 |

**\*This value is calculated from a much smaller subset of times (see N$_{ceil}$, Tables 2,3). CF abbreviations defined in Table 1.**