# Peer review of "Shallow Cumuli Cover and Its Uncertainties from Ground-based Lidar-Radar Data and Sky Images"

_Atmospheric Measurement Techniques, 2019_

## Referee Comment (RC1) · Anonymous Referee #1 · 20 Jun 2019

Dear authors, I believe your manuscript "Shallow Cumuli Cover and Its Uncertainties from Ground-based Lidar-Radar Data and Sky Images" reports on a very interesting and relevant topic which is reconciling cloud cover estimates made from various sensors during continental ShCu conditions which are known to be challenging to observe. Please find below a number of comments I have about the manuscript.

Positive points

o Figures are legible and have appropriate font size.

o Sentences are clear and properly structured.

[Figure]

o I think that the authors description of the many sensor and algorithm upgrades will add value to the literature since these are often difficult to track down and are poorly documented.

o A great novelty of this article includes comparing temporal estimates of cloud cover – from radar, ceilometer and MPL - to domain estimates of cloud cover – from TSI - trying to assess the impact of field of view.

Major comments

o Confusing uses of the terms "cloud fraction" and "cloud cover" and other derivatives

- The term "cloud fraction" is most commonly used to represent the amount of clouds present at different levels in the atmosphere and is most often presented as a profile. Here, where the authors effectively refer to the projected area of clouds at the surface, I think the term "cloud cover" would be more appropriate.

- Here, it would also seem appropriate to introduce two district cloud cover concepts: 1) "Domain cloud cover" which would be the number of cloudy pixels in each TSI image relative to the total number of pixels in each TSI image. This definition would be closest to what a large-scale model would simulate/report and is what we would ideally like to measure (i.e., "truth") and 2) "Temporal cloud cover" which is the number of cloudy pixels in time series relative to the total number of observations collected over a defined time period (e.g., in ceilometer or radar time-series over 30-min or in a TSI pixel over 15-min).

o The first goal stated by the authors is very similar to work undertaken by Kennedy et al. 2014. The authors first goal reads: "1) Have significant changes in the observations of ShCu cover occurred at the SGP site due to instrumental and algorithmic upgrades?". While I can appreciate that the current work tackles an extended dataset, to further add value, I wish it also went into more details about what are the exact algorithm changes or sensor upgrades responsible for the observed differences. For

instance:

- Given what Kennedy et al. 2014 stated about the radar: "sudden change in CF occurs around the time the radar was upgraded, suggesting that this decrease is tied to hardware sensitivity or scanning strategy changes." The current study could quantify the additional amount of cloud detected solely by the radar sensitivity increase following the change from the MMCR to the KAZR.

- Given what Kennedy et al 2014 reported about the MPL: "Addition of the MPL increases the 14-year average CF by 9 %, mainly through an increase in optically thin high clouds year-round, and mid-level clouds during the summer months." In the current study, what can the authors say about the relative importance of such a sensitivity boost relative to the number of MPL false classification for ShCu. What exact changes were implemented following 2011 to improve the MPL cloud mask and can you recommend any further algorithm modification which could help mitigate the number of false ShCu detections?

o I think the second goal of the authors should take the forefront as it tackles something that remains poorly documented in literature. As it reads in the manuscript, the authors second goal is: "2) what is the impact of FOV configurations on hourly and sub-hourly observations of ShCu cover?". Taking it a step further I would be curious to know:

- Can narrow-field of view sensors be used to estimate a cloud cover representative of a domain? and if so under what circumstances (e.g., strong horizontal wind, high cloud cover, ect.)?

I think the authors results could be used to answer this question. For instance, while the biases the authors focus on in the abstract and in the main text are between the ceilometer and TSI as estimated using a 17-year long dataset, the 1 to 1 correlation between ceilometer and TSI estimated cloud cover reported in Table B3 suggest the cloud cover statistics estimated by the two sensors converge rapidly such that ceilometer point observations could be representative of the TSI domain observations on short

timescale. The 1 to 1 correlation also improve using hourly instead of 30 min time windows. Would the authors say that this implies that hourly cloud cover estimates from ceilometer are equivalent to hourly TSI domain cloud cover? What is the optimum averaging time period where temporal cloud cover become equivalent to domain cloud cover? Would that depend on horizontal wind speed or cloud organization?

o Did the authors consider the effect of horizontal wind speed in the comparison of domain cloud cover and temporal cloud cover? I would expect that higher horizontal wind speeds advect clouds more rapidly such that, under higher wind speeds, shorter time periods of narrow-field of view measurements would be required to capture the CF observed by the wider field of view TSI.

o The spatial analysis of TSI cloud mask is very interesting but I had to read the article twice to understand where is fits in with the other cloud fraction definitions. What would make this clearer for me would be to state that the TSI "lane by lane" cloud fraction estimates are effectively temporal cloud fraction estimate (and not FOV or domain cloud fraction estimates) and that each lane can be interpreted as a time series observed by a narrow field of view sensor. I would also perhaps bring information about this lane by lane methodology and information about the radar wind profiler to section 4.3 where you describe your tool.

o Clarifications regarding the impact of insects on ShCu top detections

- Multiple studies have reported that the presence of insect hinders the radars ability to accurately detect cloud top. I think more information is needed here about how insect contamination is handled in ARSCL both pre and post 2011 where the authors hint that the MPL stopped being applied in the boundary layer. This could offer an alternative explanation to the changes in radar-lidar CF post 2011 where the increase in radar detected cloud top could be due both to the KAZR being more sensitive than the MMCR and to the KAZR insect filtering having changed such that more insect returns are misclassified as cloud tops. If both effects are in play, then I would like to

see their relative importance quantified.

o The idea of compensating bias introduced on Page 8 "introduction of compensating errors using the cloud top height criteria in the updated merged lidar-radar product." needs clarification.

- If I understand correctly the hypothesis is that in the 2000-2010 period the MPL was overly sensitive to aerosols leading to a CF overestimation while the MMCR was underly sensitive to cloud leading to a CF underestimation hence the compensating bias.

o There are gaps in the literature review

- Beyond the few studies cited on Page 2 line 30, others before have attempted to assess the representativeness/reconcile multiple cloud fraction measurements (e.g., Dr. Mariko Oue work with scanning cloud radar or Dr. Steve Schwartz work with photography or Dr. Wei Wu work with ISCC).

- Some references are missing for the bibliography (e.g., Tatarevic and Kollias, 2015)

- Some references are to meeting abstracts rather than to the published journal articles (e.g., Lamer et al. 2017 abstract work has since been published in GMD)

- Some references are miscited (e.g. Chandra et al., 2013, Zhang and Klein, 2010, 2013 and Lamer and Kollias, 2015 do not show any model-observation comparison).

o Although I understand that there are many ways to organize a methods section and that we all prefer to receive information in different sequences, for me, the layout of the data and methods section was confusing.

- I would rather the authors merge the data and methods sections which go hand in hand and preface such a section stating what quantities they are after 1) Identification of cumulus cases, which requires cloud top height and cloud cover estimates 2) Temporal cloud fraction, which will be obtained from ceilometer, ceilometer+MPL, and ceilometer+MPL+radar cloud base height time series 3) Domain cloud fraction, which

will be obtained from TSI using different angular domains 4) Providing context using horizontal wind direction and TSI lane-by-lane decomposition

o Clarification are needed when it comes to ShCu case selection

- Page 4, Line 3: Which observations are used in the ARM Shallow Cumulus data product to identify ShCu cases?

- Table 1 somewhat helps in understanding which portion of the cloud field is of interest. However, I do not see a reference to Table 1 in Sect. 3.1. where I would expect it.

- Figure 2 shows cloud fractions ranging from 0 to 1, do you consider overcast conditions to be ShCu?

- Page 4, Line 8: Why did you chose to: "additionally extending the start and end-times by 1 hour each.". This could include periods presenting deeper clouds or cloud aloft.

- Table 1. Why is a minimum cloud base height threshold applied to most CF estimates? If insects and clutter have been properly filtered from the radar data, I cannot see why this could be necessary.

- Table 1 What is the value of estimating a CFtot if the cases discussed are purely single layer ShCu? Shouldn't all cloud observed have tops and bases below ∼3 km?

o Pertinent information missing in the various definitions of cloud fraction

- Page 3 Line 15: "Appendix A contains pertinent information for their application". I believe all pertinent text should be in the main text and the appendix should be reserved for details. I am especially wanting to know how the ARSCL reports cloud top height when both the radar and MPL are used (in the 2000-2010 period) since this is very relevant to the sensitivity versus insect detection compensating effect.

Minor comments:

1) In the future, when submitting articles for review, please number the lines continu-

ously rather than restarting the numbering process on each page.

2) The abstract is very "number focused" and could benefit from including more "conclusions", for instance, the abstract does not provide information about which sensor upgrade had the largest impact on the cloud cover estimates or about the fact that cloud field organization (e.g., cloud streets) parallel to the horizontal wind direction can create large biases between narrow and wide field of view cloud cover estimates.

3) Page 1 Line 22: What is meant by "mean cloud cover"

4) Page 2 Line 2: Missing some "the"

5) Page 2 in a few places. I would suggest using the word "variability" instead of the word "changes" when referring to the cloud field

6) The acronym for the Ka-band ARM Zenith Radar should be entirely capitalized (i.e., "KAZR" not "KaZR")

7) Page 2 Line 18: What do you mean by "consistent"?

8) Page 2, Line 18: Zhang and Klein 2013 used 13 years of ARSCL data, the sentence as you have it constructed is somewhat misleading as it suggests that they used 20 years of data. It would be more appropriate to state that previous studies have used ARSCL (cite here) and this data record is now reaching 20 years in length.

9) Page 2 Line 23: Following my suggestion above "Areal cloud cover" would become domain cloud cover.

10) Page 3 Line 1: Given that the radar can be affected by insects, I would avoid using the word "reliably".

11) Page 4, Line 4: Shouldn't "ShCu cloud coverage" read "ShCu cloud periods"?

12) Figure 2 Panels a and b are missing a legend

13) Figure 2 c and d and all figures of this style are missing colorbar labels

14) Page 4, line 31: "This method has the advantages of low missing data due to multiple instruments used and limits the vertical extent of clouds." Please rephrase. Using a cloud top detection criteria does not "limits the vertical extent of clouds".

15) Page 9 line 4: Add "altitude" after "1.5 km"

16) Page 2, Line 29 "In addition, long-term averages of CF obtained from merged ceilometer-MPL data tend to be larger than FSC (Boers et al., 2010; Qian et al., 2012; Wu et al., 2014; Kennedy et al., 2014), indicating a potential consequence of instrument-dependent cloud detection differences." Could this difference not also be attributable to FOV differences? If so, please add this caveat.

17) Page 1 Line 19 "We demonstrate that CF obtained from ceilometer data alone and FSC obtained from sky images provide the most similar and consistent cloud cover estimates: bias and root-mean-square difference (RMSD) are within 0.04 and 0.12, respectively."

According to your analysis of the impact of the "Field Of View (FOV)" performed by comparing the two TSI FOV, the averaging period of the narrow field of view sensor can affect the RMSD between cloud cover obtained by the narrow and wide FOV. Am I correct to understand that this result also applies to the comparison between the ceilometer or any other "beam" observation (e.g., radar, MPL) and the TSI? If so, I think the statement above should include information about the averaging time period used for the ceilometer in this comparison.

18) Page 8 line 24: "Though a number of differences exist, the incorporation of MPL data below 3 km in the initial cloud top height retrieval algorithm between 2000-2010 but not the updated algorithm likely has a large impact (see Sect. A.4 for more details)."

I think it would help the reader to explain if an overestimation or an underestimation is expected and why?

---

## Referee Comment (RC2) · Anonymous Referee #2 · 13 Aug 2019

Review of the article titled "Shallow cumuli cover and its uncertainties from ground-based lidar-radar data and sky images" by Riley and coauthors for publication in Atmospheric Measurement Technique.

The authors have continued their work on the data collected at the ARM site during shallow cumulus cloud conditions. This seems to be a follow-on article to the Kleiss et al. 2018 article. Here the authors have compared the cloud fraction and cloud cover statistics from the vertically pointing active remote sensors (lidars and radars) with those from the sky imager. The results are robust and largely suggest that the scientists should use the TSI derived cloud statistics rather than those from the vertically

pointing instruments. The article is well-written, will be of interest to the general cloud community, and is suitable for publication in this journal. However, the article can be improved further by incorporating suggestions mentioned below.

Major suggestion

End of Page 5 and the beginning of page 6 you have mentioned the differences in the field of view of the instruments. This is good. However, the effective field of view of the radar and lidar is essentially FOV plus dwell-time times the wind speed (advection). I suggest you add few sentences to describe this effect. When you are generating the 15-min or 30-min statistics, then the differences from the two methods will largely governed by the wind speed.

In a similar vein, it will be good if you can make a figure of the CF from radar-lidar and FSC as a function of the wind speed. You already have the wind speed from the radar wind profiler, and the other two are shown in Figure 3. A figure like this will tell us how much high or low wind contributes to the differences in the two methods. Thanks.

Lastly, the lane approach is very novel. If incorporated properly, it will tell us how the clouds are organized within a cloud field and move in relation to each other. Such an analysis is outside the scope of this article. However, I suggest you add a paragraph on the potential scientific usage of the statistics derived by this approach. Thanks.

Minor Suggestions

Page 2 line 5: I think you mean "partitioning" and not "proportioning". Thanks.

Page 2 line 17: Might be better to refer to the ARM monograph in the AMS

Page 2, line 22: Remove "for example, a recent report" and just say "Zhang et al. (2017) suggested .."

Page 3, Line 2 and Line 16: Also, at other locations. Please either use radar-lidar or lidar-radar for consistency.

Page 3, line 30: "height is used here"

Page 7 line 5: Figure B1 not 1B

Figure 1 caption: I suggest using "vertical bars" rather than "error bars" to avoid confusion.

Figure 2: As the brown and blue bars are on top of each other, maybe it will be better to show them as line plots. It will be good to know how far apart they are for low CF values. Also, I see light brown bars in (a) and (b), and a dashed red line in (d). Both of these have not been explained in the caption.

Figure 3-6: It will be good if you can bin the shades in bins of 10% and use only one color for each bin. It is difficult to identify the actual values in the current versions. There are also dashed magenta lines in some of the panels.

---

## Author Comment (AC1) · 23 Oct 2019

The comment was uploaded in the form of a supplement:
https://www.atmos-meas-tech-discuss.net/amt-2019-155/amt-2019-155-AC1-supplement.pdf

---

## Author Comment (AC2) · 23 Oct 2019

The comment was uploaded in the form of a supplement:
https://www.atmos-meas-tech-discuss.net/amt-2019-155/amt-2019-155-AC2-supplement.pdf

---

## Author Comment (AC4) · 23 Oct 2019

**Shallow Cumuli Cover and Its Uncertainties from Ground-based Lidar-Radar Data and Sky Images**

Erin A. Riley1, Jessica M. Kleiss1,\*, Laura D. Riihimaki2,3, Charles N. Long2,3, Larry K. Berg4 and Evgueni Kassianov4,\*

1 Environmental Studies, Lewis and Clark College, Portland, OR 97219, USA

[revised manuscript text omitted]

---

## Author Comment (AC5) · 23 Oct 2019

Please see reviewer one response for the marked and unmarked versions of the revised manuscript for your review of the above author response. Thanks!

---

## Author Response (AR1)

Author's Response for

**Shallow Cumuli Cover and Its Uncertainties from Ground-based Lidar-Radar Data and Sky Images**
Erin A. Riley, Jessica M. Kleiss, Laura D. Riihimaki, Charles N. Long, Larry K. Berg and Evgueni Kassianov

This document contains a point-by-point response to the reviews, a list of all relevant changes made in the manuscript, and a marked-up manuscript version.

**Table of Contents**

**Response to Reviewer 1**

Thank you for the careful review of our paper and your thoughtful suggestions. We hope that you will find our responses and the corresponding revisions for the original manuscript satisfactory. Please find below your comments/suggestions (bold) and our responses with manuscript changes indicated in italic.

**Positive points**
**o Figures are legible and have appropriate font size.**
**o Sentences are clear and properly structured.**
**o I think that the authors description of the many sensor and algorithm upgrades will add value to the literature since these are often difficult to track down and are poorly documented.**
**o A great novelty of this article includes comparing temporal estimates of cloud cover – from radar, ceilometer and MPL - to domain estimates of cloud cover – from TSI trying to assess the impact of field of view.**

We are delighted that the Reviewer highlighted these positive points.

**Major Comments**:

**o 1) Confusing uses of the terms "cloud fraction" and "cloud cover" and other derivatives - The term "cloud fraction" is most commonly used to represent the amount of clouds present at different levels in the atmosphere and is most often presented as a profile. Here, where the authors effectively refer to the projected area of clouds at the surface, I think the term "cloud cover" would be more appropriate. - Here, it would also seem appropriate to introduce two district cloud cover concepts: 1)"Domain cloud cover" which would be the number of cloudy pixels in each TSI image relative to the total number of pixels in each TSI image. This definition would be closest to what a large-scale model would simulate/report and is what we would ideally like to measure (i.e., "truth") and 2) "Temporal cloud cover" which is the number of cloudy pixels in time series relative to the total number of observations collected over a defined time period (e.g., in ceilometer or radar time-series over 30-min or in a TSI pixel over 15-min).**

We agree with the Reviewer that clarifications are needed for different estimates of cloud cover. Both "cloud fraction" and "fractional sky cover" are widely accepted terms for describing cloud cover (e.g., Qian et al., 2012 and references therein), while the term "domain" frequently defines a volume with specified grid spacing and number of vertical levels for model simulations (e.g., Berg et al., 2013). Therefore, we prefer to keep these two conventional terms in our paper. To address valuable comment from the Reviewer regarding their "temporal" and "areal" representations, we have modified the abstract slightly (lines 15-18), clarified introduction of these terms in the Introduction (Sect. 1, lines 46-51), and added a reminder of terminology to the Results and Discussion (Sect. 4.2, lines 270-272).

--Abstract (lines 15-18): *Enhanced observations at this site combine the advantages of the ceilometer, micropulse lidar (MPL) and cloud radar in merged data products. Data collected by these three instruments are used to calculate narrow-FOV cloud fraction (CF) as a temporal fraction of cloudy returns within a given period. Sky images provided by TSI are used to calculate the wide-FOV fractional sky cover (FSC) as a fraction of cloudy pixels within a given image.*

--Section 1, (lines 46-51):

*There are two conventional measurement-based estimates of cloud cover: (1) cloud fraction (CF) obtained from zenith-pointing narrow-FOV observations and defined as the fraction of time when a cloud is detected within a specified period, and (2) fractional sky coverage (FSC) obtained from wide-FOV observations and defined as the fraction of cloudy pixels in a sky image. Note that FSC is similar to that estimated by a cloudy-sky observer (e.g., Henderson-Sellers and McGuffie, 1990; Kassianov et al., 2005; Long et al. 2006).*

--Section 4.2 (lines 270-272): *Recall that the CF obtained from lidar-radar observations with narrow FOV represents a transect of a cloudy sky along wind direction, while FSC acquired from wide-FOV TSI data defines an area of cloudy sky. Both the CF and the FSC are widespread measurement-based estimates of cloud cover.*

**o 2) The first goal stated by the authors is very similar to work undertaken by Kennedy et al. 2014. The authors first goal reads: "1) Have significant changes in the observations of ShCu cover occurred at the SGP site due to instrumental and algorithmic upgrades?". While I can appreciate that the current work tackles an extended dataset, to further add value, I wish it also went into more details about what are the exact algorithm changes or sensor upgrades responsible for the observed differences. For instance:**

**- Given what Kennedy et al. 2014 stated about the radar: "sudden change in CF occurs around the time the radar was upgraded, suggesting that this decrease is tied to hardware sensitivity or scanning strategy changes." The current study could quantify the additional amount of cloud detected solely by the radar sensitivity increase following the change from the MMCR to the KAZR.**

**- Given what Kennedy et al 2014 reported about the MPL: "Addition of the MPL increases the 14-year average CF by 9 %, mainly through an increase in optically thin high clouds year-round, and mid-level clouds during the summer months." In the current study, what can the authors say about the relative importance of such a sensitivity boost relative to the number of MPL false classification for ShCu. What exact changes were implemented following 2011 to improve the MPL cloud mask and can you recommend any further algorithm modification which could help mitigate the number of false ShCu detections?**

The Reviewer is right that the algorithm changes and/or sensor upgrades are interesting topics, but their detailed discussion is beyond the scope of our current work.

However, we have expanded and clarified several important issues associated with these important topics.

- The radar-related changes and upgrades: Appendix A summarizes the instrumental and algorithmic changes of the radar. The primary limitation of the radar is the difficulty of observing optically thin clouds with low liquid water content (LWC), such as continental ShCu. The radar upgrades have primarily decreased the dwell time resulting in an increased number of pulses averaged. However, these upgrades very likely have not improved detection of the optically thin clouds. We have added a clarification regarding the sensitivity of the radar to cloud optical depth. (Section A.2, lines 463-465; lines 477-479).

- The MPL-related changes: The original cloud mask involved a signal-to-noise threshold based on cloud droplet scattering *(Campbell et al. 1998)*, whereas the updated cloud mask incorporated information on different scattering properties of aerosol and clouds (e.g., Wang and Sussen 2001). The corresponding clarifications have been added  (Section 4.1; lines 254-258) and (Section A.3, Lines 481-483).

The specific changes are documented below.

--Section A.2 (lines 463-465):
*Many continental cumuli are optically thin (liquid water content less than 50 g/m2) and have small droplets. As a result, the cloud radars (both MMCR and KAZR) are not able to "see" the majority of these clouds (e.g., Chandra et al. 2013; Lamer and Kollias, 2015).*

--Section A.2 (lines 477-479)
Cloud radars (both the MMCR and KAZR) are known to miss detection of ShCu with small droplets compared to the ceilometer. For example, Chandra et al (2013) have found that the MMCR misses the majority of continental ShCu observed at the SGP site.

--Section 4.1 (lines 254-258): *Though a number of differences exist, the incorporation of MPL data (below 3 km) in the original cloud top height retrieval would increase number of detected cloud tops compared to those retrieved from the radar data alone for the initial period (2000-2010). Reliance only on the radar data for cloud top detection in the updated algorithm would result in fewer cloud top height detections and therefore a lower CF (see Sect. A.4 for more details).*

--Section A.3 (lines 481-483):
*The original cloud mask involved a signal-to-noise threshold based on cloud droplet scattering (Campbell et al. 1998), whereas the updated cloud mask incorporated information on different scattering properties of aerosol and clouds (Wang and Sussen 2001). … It is expected that the updated cloud mask would improve cloud returns.*

**O 3) I think the second goal of the authors should take the forefront as it tackles something that remains poorly documented in literature. As it reads in the manuscript, the authors second goal is: "2) what is the impact of FOV configurations on hourly and sub-hourly observations of ShCu cover?". Taking it a step further I would be curious to know:**
**- Can narrow-field of view sensors be used to estimate a cloud cover representative of a domain?**
**- if so under what circumstances (e.g., strong horizontal wind, high cloud cover, ect.)?**

In contrast to simulation-based studies, we use observations to quantify the impact of instrumental FOVs on cloud cover estimates. However, we should clarify that areas "seen" by ground-based instruments with narrow- and wide-FOV at the cloud base height are much smaller than areas (~ 30 km) considered in simulation-based studies (e.g., Oue et al. 2016)  For example, TSI with 100-deg FOV "sees" a moderate area (~ 2.4 km diameter) at typical cloud based height (1km).

A clarification of differences between our approach and previous studies is included in section 4.2 (Lines 278-284):

*In particular, the previous model studies (Astin et al., 2001; Berg and Stull, 2002) have demonstrated that the cloud cover obtained from the transect measurements mimics the area-averaged cloud cover for non-organized (e.g. random) cloud fields well if the sample size is relatively large (or numerous individual clouds are sampled). Recently Oue et al. (2016) have showed that 10 or more ceilometers equally spaced across a 25 km width in the cross-wind direction are required to estimate the simulated cloud cover in the small (30 km) domain. Certainly, the number of ceilometers, their locations and averaging time required for an accurate estimation of the cloud cover depend on the spatial arrangement of clouds and wind speed.*

With this clarification in mind, we respond to the Reviewer's questions by explicitly stating that we interpret the term "domain" in the question to be an area "seen" by TSI with the 100-deg FOV at the cloud base height.

1) **Can narrow-field of view sensors be used to estimate a cloud cover representative of a domain?**
Our results (Section 4.1) suggest that a narrow-FOV data from ceilometer as compared with those from lidar and radar can be used to estimate wide-FOV FSC reasonably well, on average, in terms of bias (Table 3), slope and intercept of linear regression (Table B2), and correlation coefficient (Table B3). For example, the correlation coefficient values for sub-hourly and hourly time scales are 0.83 and 0.89, respectively (Table B3). It should be emphasized that about 30-34% of the corresponding CF-FSC comparisons still have difference greater than 0.1 (section 4.2, lines 301-303). The level of agreement between the narrow-FOV CF and wide-FOV FSC depends on several factors described below (the second question).

2) **if so under what circumstances (e.g., strong horizontal wind, high cloud cover, ect.)?**
Several factors, such as wind speed and spatial arrangement of clouds, determine whether or not the narrow-FOV CF and wide-FOV FSC are comparable, on average. The spatial arrangement of clouds (e.g., organized versus non-organized spatial distribution) defines representation of a 1D transect along a wind direction of a 2D cloud field for a given area of interest, while the wind speed determines the number of sampled clouds (or sample size). The "quick-look" tool (Section 4.3) can be used for characterization of the spatial arrangement of clouds while available information on the wind speed can be used for estimation of the sample size, and therefore for assessment of the wind speed impact on the CF-FSC comparison.

To address these points in the manuscript, two new paragraphs have been added. The first new paragraph discusses possible applications of the "quick-look" tool to assess the impact of cloud field organization on agreement between narrow- and wide-FOV observations (Section 4.3, lines 365-375). The second new paragraph describes the impact of wind speed on the agreement of narrow- and wide-FOV observations (Section 4.2, lines 306-3017). A new Figure 6 has been added to illustrate the concepts discussed in the second new paragraph.

The new paragraph discussing the impact of cloud field organization on the agreement of narrow- and wide-FOV observations (Section 4.3, lines 365-375):

*"There are two main expected applications of the introduced "quick-look" tool. The first potential application is a classification of spatial organization of cloud fields using, for example, cross-wind cloud field variability (e.g. peaks and valleys in Fig. 7b) and within-lane variance of cloud amount (e.g. vertical bars in Fig. 7b). Numerous images generated by the "quick-look" tool (e.g., Figure 8b) for the extended period (2000-2017) can be considered as a valuable training dataset for machine learning with focus on automated detection of desired features of the cloud fields (e.g., "cloud streets") and unwanted contaminations of TSI images (e.g., Figure 9). Second potential application is a visual inspection of the generated images for a given period of interest (e.g., a short-term field campaign) to check for the impact of instrumental detection differences and cloud field organization on the observed cloud amount. Visual inspection may be feasible given a limited number (about 40) of ShCu events annually during the warm season. For example, a spread of the lane CFs (gray region in Fig. 8c) gives an idea about the cross-wind cloud field variability within a given FOV, and thus aids in understanding the difference between cloud amounts obtained from the narrow- and wide-FOV observations."*

The new paragraph discussing the impact of wind speed and the sample size on the agreement of narrow- and wide-FOV observations (Section 4.2, lines 306-317).

*"The effective spatial area sampled by either narrow or wide FOV instruments is a function of both sampling duration and wind speed. High wind speed in comparison with low wind speed (1) increases sample size for a given period and (2) tends to organize horizontal arrangement of clouds (e.g., Weckworth et al. 1999, Atkinson and Zhang 1996). These two factors associated with sample size and spatial arrangement of clouds should be considered when differences between cloud cover obtained from narrow- and wide-FOV observations as function of wind speed are considered (Fig. 6). In particular, Figure 6 illustrates that both CF-FSC and "CF-like"-FSC differences are reduced noticeably as the wind speed increases from 1 m/s to 3 m/s, and continue to reduce slightly as the wind speed grows up to 11 m/s. The CF-FSC and "CF-like"-FSC differences obtained at a higher wind speed (above 11 m/s) should be considered with caution due to limited number of the corresponding cases with high wind speed (e.g., fewer than 100 cases for 60-min time average). The increased sampling area associated with increased wind speed does not necessarily result in an improved agreement between the narrow- and wide-FOV observations for both hourly and sub-hourly observations due to the impact of wind speed on cloud organization."*

**o 4) Did the authors consider the effect of horizontal wind speed in the comparison of domain cloud cover and temporal cloud cover? I would expect that higher horizontal wind speeds advect clouds more rapidly such that, under higher wind speeds, shorter time periods of narrow-field of view measurements would be required to capture the CF observed by the wider field of view TSI.**

To address your important questions, a new paragraph regarding the wind speed and the sample size (Sect. 4.2, lines 306-317) together with a new plot (Figure 6) have been added. This paragraph has been included in our reply above.

**o 5) The spatial analysis of TSI cloud mask is very interesting but I had to read the article twice to understand where is fits in with the other cloud fraction definitions. What would make this clearer for me would be to state that the TSI "lane by lane" cloud fraction estimates are effectively temporal cloud fraction estimate (and not FOV or domain cloud fraction estimates) and that each lane can be interpreted as a time series observed by a narrow field of view sensor. I would also perhaps bring information about this lane by lane methodology and information about the radar wind profiler to section 4.3 where you describe your tool.**

Thank you for the valuable suggestions. We have moved the description of the lane-by-lane methodology and the radar wind profiler (previous section 3.5) to section 4.3, added clarification to section 4.3, and changed the labels on Figure 7 from FSC to CF.

The clarification (Section 4.3; lines 328-330)
*Each pixel in the averaged image can be interpreted as a 15-min CF measurement from a narrow-FOV sensor. The variability of CF in the cross-wind direction can indicate the possible influence of cloud field organization on cloud cover estimates provided by narrow-FOV observations.*

Also, the vertical axis label for Figure 7b has been changed from FSC to CF for consistency with the above changes.

**o 6) Clarifications regarding the impact of insects on ShCu top detections - Multiple studies have reported that the presence of insect hinders the radars ability to accurately detect cloud top. I think more information is needed here about how insect contamination is handled in ARSCL both pre and post 2011 where the authors hint that the MPL stopped being applied in the boundary layer. This could offer an alternative explanation to the changes in radar-lidar CF post 2011 where**

*the increase in radar detected cloud top could be due both to the KAZR being more sensitive than the MMCR and to the KAZR insect filtering having changed such that more insect returns are misclassified as cloud tops. If both effects are in play, then I would like to see their relative importance quantified.*

We agree that insects can contaminate accurate determination of cloud boundaries by radar. However, accurate cloud top height retrievals by radar is not required in our analysis because a simple threshold is used to determine the presence of ShCu. Moreover, cloud base height estimation involves lidar observations (both ceilometer and MPL) which are not impacted by the presence of insects,

Text has been added to clarify this point: (Section 3.1, lines 141-144)
*Insect contamination may contribute to significant uncertainty of the radar-based retrievals of cloud boundaries. Therefore, our analysis employs a semi-quantitative threshold approach when using the cloud top heights. This approach is less sensitive to the insect contamination.*

**o 7) The idea of compensating bias introduced on Page 8 "introduction of compensating errors using the cloud top height criteria in the updated merged lidar-radar product." needs clarification.**
**- If I understand correctly the hypothesis is that in the 2000-2010 period the MPL was overly sensitive to aerosols leading to a CF overestimation while the MMCR was underly sensitive to cloud leading to a CF underestimation hence the compensating bias**

Thank you for pointing this out. For the later sub-period (2010-2017), the merged cloud radar-lidar product relies on the "shallow" (< 3 km) radar data instead of the combined MPL-radar observations for determining the cloud top. The radar misses a substantial fraction (about 30%) of ShCu, therefore the cloud top height (below 3 km) is very likely to be missed. Meanwhile, the merged lidars (ceilometer and MPL) data are used to detect the cloud base height and exhibit higher CF than that from the ceilometer alone. A compensating error could potentially arise from the over-detection of clouds in the merged lidar data with the under-detection of cloud from the radar observations. The RMSD for the CF including cloud top heights for the later sub-period (2010-2017) is higher than those for the CF obtained from ceilometer alone (even for near-zero bias). This indicates that the instrument detection differences in the merged lidar-radar product contribute mostly to the CF uncertainty.

**o There are gaps in the literature review**
**- Beyond the few studies cited on Page 2 line 30, others before have attempted to assess the representativeness/reconcile multiple cloud fraction measurements (e.g., Dr. Mariko Oue work with scanning cloud radar or Dr. Steve Schwartz work with photography or Dr. Wei Wu work with ISCC).**
**- Some references are missing for the bibliography (e.g., Tatarevic and Kollias, 2015)**
**- Some references are to meeting abstracts rather than to the published journal articles (e.g., Lamer et al. 2017 abstract work has since been published in GMD)**
**- Some references are miscited (e.g. Chandra et al., 2013, Zhang and Klein, 2010, 2013 and Lamer and Kollias, 2015 do not show any model-observation comparison).**

We thank the Reviewer for the careful editing and helpful suggestions. We have made the following adjustments to the manuscript.

Introduction (Lines 69-71)
- We have added Oue et al. (2016) reference along with two citations in the text.

*Moreover, sampling of LES-generated cloud fields by a virtual instrument can be a helpful way to reconcile debated differences between the retrieved and predicted values of cloud cover (Oue et al., 2016).*

Section 4.2 (lines 278-284)

*In particular, the previous model studies (Astin et al., 2001; Berg and Stull, 2002) have demonstrated that the cloud cover obtained from the transect measurements mimics the area-averaged cloud cover for non-organized (e.g. random) cloud fields well if the sample size is relatively large (or numerous individual clouds are sampled). Recently Oue et al. (2016) have showed that 10 or more ceilometers equally spaced across a 25 km width in the cross-wind direction are required to estimate the simulated cloud cover in the small (30 km) domain. Certainly, the number of ceilometers, their locations and averaging time required for an accurate estimation of the cloud cover depend on the spatial arrangement of clouds and wind speed.*

- We have added Tatarevic and Kollias (2015) to the bibliography,

Conclusions (line 408)
We have replaced the Lamer et al. (2017) reference with Lamer et al. (2018) GMD article, and adjusted the bibliography accordingly.

Introduction (lines 52-53)
- We have removed the citations to Chandra et al., 2013, Zhang and Klein, 2010, 2013 and Lamer and Kollias, 2015 for model-observation comparison, and inserted Zhang et al., 2017 and Endo et al., 2019.

**o 9) Although I understand that there are many ways to organize a methods section and that we all prefer to receive information in different sequences, for me, the layout of the data and methods section was confusing.**
**- I would rather the authors merge the data and methods sections which go hand in hand and preface such a section stating what quantities they are after 1) Identification of cumulus cases, which requires cloud top height and cloud cover estimates 2) Temporal cloud fraction, which will be obtained from ceilometer, ceilometer+MPL, and ceilometer+MPL+radar cloud base height time series 3) Domain cloud fraction, which will be obtained from TSI using different angular domains 4) Providing context using horizontal wind direction and TSI lane-by-lane decomposition**

We thank the Reviewer for this helpful comment. Indeed, during development of this manuscript we have attempted many possible organizational approaches to communicating the data and methods used to produce this dataset. One challenge with the approach that the Reviewer suggests is the interdependent nature of these data. For example, the natural first quantity of the paper, as you list above, is the identification of ShCu events. However, this method requires TSI, ceilometer, and lidar/radar data (items #2 and 3 above). Thus we settled on the current organizational structure of the data and methods sections.

While the current layout may be somewhat cumbersome to read as prose, we believe this structure will be most conducive for readers to access the data and perform their own analyses in extension of this work.

**o 10) Clarification are needed when it comes to ShCu case selection**

Thank you for your careful attention to detail in the review of this paper. We address each point independently below.

**- Page 4, Line 3: Which observations are used in the ARM Shallow Cumulus data product to identify ShCu cases?**

The reference to the new Shallow Cumulus data product has been added for clarification of the observations used in our study. (Section 2, lines 108-110)
*The newly released ARM Shallow Cumulus data product (Data Reference: Shi et al., 2000) identifies times of ShCu from lidar / radar cloud boundary heights and includes FSC from TSI observations (Lim et al., 2018).*

**- Table 1 somewhat helps in understanding which portion of the cloud field is of interest. However, I do not see a reference to Table 1 in Sect. 3.1. where I would expect it.**

We respectfully note that Table 1 is referenced in the third sentence in Sect. 3.1.  To facilitate readability, we have added another reference to it later in this paragraph. (Line 146)

**- Figure 2 shows cloud fractions ranging from 0 to 1, do you consider overcast conditions to be ShCu?**

The FSC obtained from 100-deg FOV TSI data was used to determine clear-sky or overcast conditions. Only partly-cloudy conditions (0.05<FSC<0.95) were considered in our study. This is stated in Section 3.4 (line 185), and the captions of Figure 2, Table B1, Table 2, Table 3, and Figure B1.

**- Page 4, Line 8: Why did you chose to: "additionally extending the start and end-times by 1 hour each.". This could include periods presenting deeper clouds or cloud aloft.**

The text describing Shallow Cumulus event detection was revised to improve clarity. (Section 2, lines 134-138)
*The extension allows for more accurate determination of the start- and end- times of the event on the finer time scale of the TSI FSC (15 min). Quality control procedures (Sect. 3.4) are used to censor multi-layer clouds and clear sky conditions on the 15-min and hourly observations of cloud cover.*

**- Table 1. Why is a minimum cloud base height threshold applied to most CF estimates? If insects and clutter have been properly filtered from the radar data, I cannot see why this could be necessary.**

This helps additionally screen out smoke from agricultural burning of biomass, which happens several days each warm season. It would eliminate fog too, which has never been an issue.

**- Table 1 What is the value of estimating a CFtot if the cases discussed are purely single layer ShCu? Shouldn't all cloud observed have tops and bases below 3 km?**

The ShCu data product includes ShCu cases with overlying cirrus. The TSI cannot differentiate low clouds from  cirrus, which can be sporadic with short duration. The selected cases were required to have at least 2 hours without cirrus.

**o 11) Pertinent information missing in the various definitions of cloud fraction**

**- Page 3 Line 15: "Appendix A contains pertinent information for their application". I believe all pertinent text should be in the main text and the appendix should be reserved for details. I am especially wanting to know how the ARSCL reports cloud top height when both the radar and MPL are used (in the 2000-2010 period) since this is very relevant to the sensitivity versus insect detection compensating effect.**

Appendix A provides detailed information about data streams and data pre-processing. This is valuable background knowledge for researchers.

The issue of cloud top height observations in the earlier (2000-2010) and later (2011-2017) periods is indeed interesting and relevant and discussed in full in the Conclusions. In particular, the corresponding discussion under the first research question "*1) Have significant changes in the observations of ShCu cover occurred at the SGP site due to instrumental and algorithmic upgrades?*" is included*. (lines 383-394)

**Minor Comments**:

**1) In the future, when submitting articles for review, please number the lines continuously rather than restarting the numbering process on each page.**

We agree that continuous line numbering is preferable - The revised manuscript uses continuous line numbering.

**2) The abstract is very "number focused" and could benefit from including more "conclusions", for instance, the abstract does not provide information about which sensor upgrade had the largest impact on the cloud cover estimates or about the fact that cloud field organization (e.g., cloud streets) parallel to the horizontal wind direction can create large biases between narrow and wide field of view cloud cover estimates.**

We have addressed the first point by adding a statement to the abstract regarding the algorithmic upgrade that had the largest impact on the cloud cover estimate (lines 26-27), and we have addressed the second point regarding the potential bias in observations due to cloud field organization (lines 32-33). The specific edited text is below.

Abstract (lines 26-27): *The strong period dependence of CF obtained from the combined ceilometer-MPL-radar data is likely due to increased reliance on the radar for cloud top height returns.*

Abstract (lines 32-33): *The influence of cloud field organization, such cloud streets parallel to the wind direction, on narrow- and wide-FOV cloud cover estimates can be visually assessed.*

**3) Page 1 Line 22: What is meant by "mean cloud cover"**

The term "mean cloud cover" has been replaced with the term "multi-year mean cloud cover" (Abstract, line 23)

**4) Page 2 Line 2: Missing some "the"**

Thank you - this has been changed. (Section 1, line 40)

**5) Page 2 in a few places. I would suggest using the word "variability" instead of the word "changes" when referring to the cloud field**

Thank you - this has been changed in the Introduction, lines 43 and 45.

**6) The acronym for the Ka-band ARM Zenith Radar should be entirely capitalized (i.e., "KAZR" not "KaZR")**

Thank you - all instances have been changed.

**7) Page 2 Line 18: What do you mean by "consistent"?**

The word "consistent" has been removed to avoid confusion. (Page 2, line 56)

**8) Page 2, Line 18: Zhang and Klein 2013 used 13 years of ARSCL data, the sentence as you have it constructed is somewhat misleading as it suggests that they used 20 years of data. It would be more appropriate to state that previous studies have used ARSCL (cite here) and this data record is now reaching 20 years in length.**

The text has been changed to indicate that the ARSCL data have been available since Nov. 1996 to avoid confusion. The revised sentence reads:

(Introduction, lines 56-57) *A merged lidar-radar data product is available from Nov. 1996 to the present at the SGP site and has served as a basis for developing ShCu climatology at the SGP site …*

**9) Page 2 Line 23: Following my suggestion above "Areal cloud cover" would become domain cloud cover.**

We've elected to maintain our conventions. Please see the discussion in the first question of this review. Thank you for your suggestion.

**10) Page 5 Line 1: Given that the radar can be affected by insects, I would avoid using the word "reliably".**

Thank you for this suggestion. The wording of these sentences was changed in response to major comment number 6.

**11) Page 4, Line 4: Shouldn't "ShCu cloud coverage" read "ShCu cloud periods"?**

Yes, thank you. This has been changed to *times of ShCu*. (Section 2, line 109)

**12) Figure 2 Panels a and b are missing a legend**

Note that this is now Figure 1. The caption has been updated. Note that the 1D histograms were changed to line plots in response to a separate Reviewer comment.

Figure 1 (caption): *merged ceilometer-MPL (blue) and merged lidar-radar (red)*

**13) Figure 2 c and d and all figures of this style are missing colorbar labels**

Thank you. The captions for figures 1, has been updated to include specific mention of the colorbar.

Figure 1 (caption): *joint histograms (counts)*
Figure 2 (caption): *Color scale represents counts in increments of 10.*

**14) Page 4, line 31: "This method has the advantages of low missing data due to multiple instruments used and limits the vertical extent of clouds." Please rephrase. Using a cloud top detection criteria does not "limits the vertical extent of clouds".**

Thank you for this suggested clarification. The relevant passage has been changed:
(Section 3.1, lines 148-149) *This method has the advantages of low missing data due to use of multiple instruments and incorporates information about cloud top height consistent with the definition of shallow convection.*

**15) Page 9 line 4: Add "altitude" after "1.5 km"**
We respectfully note that the sentence already reads "*1.5 km cloud base*", which employs the correct terminology. (Section 4.2, line 272)

**16) Page 2, Line 29 "In addition, long-term averages of CF obtained from merged ceilometer-MPL data tend to be larger than FSC (Boers et al., 2010; Qian et al., 2012; Wu et al., 2014; Kennedy et al., 2014), indicating a potential consequence of instrument-dependent cloud detection differences." Could this difference not also be attributable to FOV differences? If so, please add this caveat.**

Long-term averages are monthly and yearly long, and have sufficient sample size that reduce the impacts of limited FOV. A potential long-term impact of FOV configuration on cloud cover estimates could be associated with cloud sides visible at moderate zenith (viewing) angles. However, one would then expect that wide-FOV FSC would exceed narrow-FOV CF. The opposite is demonstrated by our analysis. This suggests that the obtained difference between narrow-FOV CF and wide-FOV FSC is mainly due to instrument-dependent cloud detection differences. (revised manuscript Page 3, line 67)

**17) Page 1 Line 19 "We demonstrate that CF obtained from ceilometer data alone and FSC obtained from sky images provide the most similar and consistent cloud cover estimates: bias and root-mean-square difference (RMSD) are within 0.04 and 0.12, respectively."**
**According to your analysis of the impact of the "Field Of View (FOV)" performed by comparing the two TSI FOV, the averaging period of the narrow field of view sensor can affect the RMSD between cloud cover obtained by the narrow and wide FOV. Am I correct to understand that this result also applies to the comparison between the ceilometer or any other "beam" observation (e.g., radar, MPL) and the TSI? If so, I think the statement above should include information about the averaging time period used for the ceilometer in this comparison.**

Thank you for the careful attention to detail. Yes, the "CF-like" aims to characterize the effect of FOV on cloud amount observations, since it is derived from the same instrument (the TSI) as the wide-FOV FSC. Indeed, the characterization of the RMSD between "CF-like" and FSC is 0.1 for 60-minute averages, and 0.15 for fine temporal scale (30-minute CF-like, 15-minute FSC). We accept your suggestion, and edit the text to read as follows:

(Abstract, lines 21-23) *We demonstrate that CF obtained from ceilometer data alone and FSC obtained from sky images provide the most similar and consistent cloud cover estimates: hourly bias and root-mean-square difference (RMSD) are within 0.04 and 0.12, respectively.*

**18) Page 8 line 24: "Though a number of differences exist, the incorporation of MPL data below 3 km in the initial cloud top height retrieval algorithm between 2000-2010 but not the updated algorithm likely has a large impact (see Sect. A.4 for more details)."**
**I think it would help the reader to explain if an overestimation or an underestimation is expected and why?**

This sentence has been changed to read:

(Section 4.1, lines 253-257) *Though a number of differences exist, the incorporation of MPL data below 3 km in the original cloud top height retrieval would increase the number of detected cloud tops compared to those retrieved from the radar data alone for the initial period (2000-2010). Reliance only on the radar data for cloud top detection in the updated algorithm would result in fewer cloud top height detections and therefore a lower CF (see Sect. A.4 for more details).*

**Response to Reviewer 2**

Thank you for the careful review of our paper and your thoughtful suggestions. We hope that you will find our responses and the corresponding revisions for the original manuscript satisfactory. Please find below your comments/suggestions (bold) and our responses with manuscript changes indicated in italic.

**Major suggestions**

**1. End of Page 5 and the beginning of page 6 you have mentioned the differences in the field of view of the instruments. This is good. However, the effective field of view of the radar and lidar is essentially FOV plus dwell-time times the wind speed (advection). I suggest you add a few sentences to describe this effect. When you are generating the 15-min or 30-min statistics, then the differences from the two methods will largely governed by the wind speed.**

We believe your comment is related to narrow-FOV FSC ("CF-like") introduced in Section 3.3. We've added the following three sentences (section 3.3, lines 174-78):

*The narrow-FOV (3x3 pixel region) "CF-like" observation has 4.1° (original) and 2.3°(improved) angular resolution. For comparison, the narrow-FOV ARSCL cloud products, including the lidar-radar CF, have much finer angular resolution (about 0.2°). The corresponding spatial resolution of the lidar-radar CF can be estimated by multiplying wind speed at cloud base height by lidar-radar dwell time (about 10 sec). For example, its spatial resolution is about 100 m for 10 m/sec wind speed.*

**2. In a similar vein, it will be good if you can make a figure of the CF from radar-lidar and FSC as a function of the wind speed. You already have the wind speed from the radar wind profiler, and the other two are shown in Figure 3. A figure like this will tell us how much high or low wind contributes to the differences in the two methods. Thanks.**

As you suggested, we've added a new Figure 6 in the revised manuscript to illustrate relative contributions of low and high wind speed to the difference between cloud cover obtained from the narrow- and wide-FOV observations. Also, we've added the following paragraph (section 4.2, lines 305-316).

*The effective spatial area sampled by either narrow or wide FOV instruments is a function of both sampling duration and wind speed. High wind speed in comparison with low wind speed (1) increases sample size for a given period and (2) tends to organize horizontal arrangement of clouds (e.g., Weckworth et al. 1999, Atkinson and Zhang 1996). These two factors associated with sample size and spatial arrangement of clouds should be considered when differences between cloud cover obtained from narrow- and wide-FOV observations as function of wind speed are considered (Fig. 6). In particular, Figure 6 illustrates that both CF-FSC and "CF-like"-FSC differences are reduced noticeably as the wind speed increases from 1 m/s to 3 m/s, and*

*continue to reduce slightly as the wind speed grows up to 11 m/s. The CF-FSC and "CF-like"-FSC differences obtained at a higher wind speed (above 11 m/s) should be considered with caution due to limited number of the corresponding cases with high wind speed (e.g., fewer than 100 cases for 60-min time average). The increased sampling area associated with increased wind speed does not necessarily result in an improved agreement between the narrow- and wide-FOV observations for both hourly and sub-hourly observations due to the impact of wind speed on cloud organization.*

**3. Lastly, the lane approach is very novel. If incorporated properly, it will tell us how the clouds are organized within a cloud field and move in relation to each other. Such an analysis is outside the scope of this article. However, I suggest you add a paragraph on the potential scientific usage of the statistics derived by this approach. Thanks.**

Thank you for your valuable suggestion. We've added a new paragraph (section 4.3, lines 464-374) in the revised manuscript to highlight expected scientific applications of our "quick-look" tool.

*There are two main expected applications of the introduced "quick-look" tool. The first potential application is a classification of spatial organization of cloud fields using, for example, cross-wind cloud field variability (e.g. peaks and valleys in Fig. 7b) and within-lane variance of cloud amount (e.g. vertical bars in Fig. 7b). Numerous images generated by the "quick-look" tool (e.g., Figure 8b) for the extended period (2000-2017) can be considered as a valuable training dataset for machine learning with focus on automated detection of desired features of the cloud fields (e.g., "cloud streets") and unwanted contaminations of TSI images (e.g., Figure 9). Second potential application is a visual inspection of the generated images for a given period of interest (e.g., a short-term field campaign) to check for the impact of instrumental detection differences and cloud field organization on the observed cloud amount. Visual inspection may be feasible given a limited number (about 40) of ShCu events annually during the warm season. For example, a spread of the lane CFs (gray region in Fig. 8c) gives an idea about the cross-wind cloud field variability within a given FOV, and thus aids in understanding the difference between cloud amounts obtained from the narrow- and wide-FOV observations.*

**Minor Suggestions**

**Page 2 line 5: I think you mean "partitioning" and not "proportioning". Thanks**.
- Corrected, thank you.

**Page 2 line 17: Might be better to refer to the ARM monograph in the AMS**
- Citation to the ARM Program website (https://www.arm.gov/) was added.

**Page 2, line 22: Remove "for example, a recent report" and just say "Zhang et al. (2017) suggested .."**
- Corrected, thank you.

**Page 3, Line 2 and Line 16: Also, at other locations. Please either use radar-lidar or lidar-radar for consistency.**
- Changed to lidar-radar throughout.

**Page 3, line 30: "height is used here"**
- Corrected, thank you.

**Page 7 line 5: Figure B1 not 1B**
- Figure 1a and 1b are now Figure 7a and 7b in the revised manuscript.
- All figure names have been changed as a result of inserting new Figure 6.

**Figure 1 caption: I suggest using "vertical bars" rather than "error bars" to avoid confusion.**
- Suggestion accepted, thank you.

**Figure 2: As the brown and blue bars are on top of each other, maybe it will be better to show them as line plots. It will be good to know how far apart they are for low CF values. Also, I see light brown bars in (a) and (b), and a dashed red line in (d). Both of these have not been explained in the caption.**
- Figure 2 is now Figure 1 in the revised manuscript.
- Thanks for the suggestion. This figure has been updated to use a line plot instead of bars. The figure caption has been edited.

**Figure 3-6: It will be good if you can bin the shades in bins of 10% and use only one color for each bin. It is difficult to identify the actual values in the current versions. There are also dashed magenta lines in some of the panels.**
- These are count histograms. We've changed the colorbar to be in increments of 10 counts, which seems to improve the legibility.
- We have clarified that the units are "counts" in the legends of figures 1 and 2.

**List of all relevant changes made in the manuscript**

- The Abstract has been clarified to emphasize the qualitative (as well as quantitative) findings from this study.
- The text has been clarified to differentiate between the wide field of view (FOV) observations from the total sky imager (TSI) and the narrow-FOV observations from active remote sensing (lidar / radar).
- The description of the process to determine times of shallow cumulus was clarified.
- The possible contamination by insects on radar data in the context of this study was included
- An expanded discussion is included on the impact of insects on the radar retrievals of cloud top height, and benefits of the merged lidar/radar product for cloud top height estimates.
- The influence of wind at cloud base height on the effective field of view of narrow remote sensors was included.
- The methodology to perform the novel spatial analysis of wide-FOV sky images is moved from the methods section to the Results and Discussion section, immediately before the implications of this analysis are presented.
- A new discussion and new figure 6 are added to address the influence of wind at cloud base on the effective field of view of both narrow- and wide-FOV observations.
- A discussion is added to indicate possible applications and future uses of the spatial analysis "quick look" tool.
- Additional technical details on the algorithm changes and instrumental upgrades for the radar and lidar are included in the appendices.

- Some stylistic changes to the text have been applied throughout, including consistent use of the acronym KAZR, and "lidar/radar" to indicate active remote sensing.
- 8 additional references were added describing previous work that is relevant to this study.

- Figures have been re-ordered to present the instrumental and time-averaging differences prior to introduction of the spatial analysis "quick look" tool.
- Figure color bars and line types have been adjusted for increased clarity.

[revised manuscript text omitted]

---

## Referee Report (RR1)

Dear authors, thank you for taking the time to address my comments. After reading your responses, I have only a few outstanding questions.
* * *
*New reviewers' comment:*

One of the main conclusions of this article is "Whereas CF from merged MPL-ceilometer data provides the largest estimates of the multi-year mean cloud cover: about 0.12 (35%) and 0.08 (24%) greater than FSC for the first and second sub-periods, respectively. CF from merged ceilometer-MPL-radar data has the strongest sub-period dependence with a bias of 0.08 (24%) compared to FSC for the first sub-period and shows no bias for the second sub-period. The strong period dependence of CF obtained from the combined ceilometer-MPL-radar data is likely due to increased reliance on the radar for cloud top height returns."

This statement leaves me with the following thoughts:

1) If the largest factor affect the ceilometer-MPL-radar CF estimate is whether or not the radar detects a cloud top then I think the authors should reconsider their answer to my original comment about the role of insects.

*Reviewers original comment:* "Clarifications regarding the impact of insects on ShCu top detections - Multiple studies have reported that the presence of insect hinders the radars ability to accurately detect cloud top. I think more information is needed here about how insect contamination is handled in ARSCL both pre and post 2011 where the authors hint that the MPL stopped being applied in the boundary layer. This could offer an alternative explanation to the changes in radar-lidar CF post 2011 where the increase in radar detected cloud top could be due both to the KAZR being more sensitive than the MMCR and to the KAZR insect filtering having changed such that more insect returns are misclassified as cloud tops. If both effects are in play, then I would like to see their relative importance quantified."

> Let me be clearer about this statement; By "accurately" I meant that sometimes the radar cannot detect the "real" cloud at all but can detect an insect layer which happens to fly above a lidar-detected cloud base. This would lead, for the same "real" cloud, to CF only being estimated by the ceilometer-MPL-radar algorithm if insects are detected and interpreted as being cloud; thus generating biases if the insect filter is altered.

*Authors original response:* "We agree that insects can contaminate accurate determination of cloud boundaries by radar. However, accurate cloud top height retrievals by radar is not required in our analysis because a simple threshold is used to determine the presence of ShCu. Moreover, cloud base height estimation involves lidar observations (both ceilometer and MPL) which are not impacted by the presence of insects, Text has been added to clarify this point: (Section 3.1, lines 141-144) "Insect contamination may contribute to significant uncertainty of the radar-based retrievals of cloud boundaries. Therefore, our analysis employs a semi-quantitative threshold approach when using the cloud top heights. This approach is less sensitive to the insect contamination."

> I agree with the authors that "cloud base height estimation involves lidar observations (both ceilometer and MPL) which are not impacted by the presence of insects". But the importance of detecting "true" cloud tops should not be undermined since it is a fundamental criterion in the ceilometer-MPL-radar CF estimate. Given this, I think the authors should specify how insect filtering is performed both pre and post 2011. I understand that this may have been the topic of

previous studies but it would make this manuscript much more thorough to statement the main points of these algorithms; For instance, are they relying on polarimetric variables, are they threshold based, do they have continuity arguments? I would think something has changed after the much improved KAZR was installed post-2011. I also think the potential bias insects could generate should be specifically stated in the conclusions perhaps at the end of point 1).

2) What does "increased reliance" mean. Going to the conclusion section I get a sense that "increased reliance" means only relying on the radar for cloud top detection below 3km. I would suggest taking the following statement out of the parenthesis in the conclusions "(MPL is not used below 3km in the second sub-period)" since it is a very important part of the sentence. Also, I would suggest making a small change to the abstract to improve clarity: "The strong period dependence of CF obtained from the combined ceilometer-MPL-radar data likely results from a change in what sensors are relied on the detect clouds below 3km; Post 2011, the MPL stopped being used for cloud detection below 3km leaving the radar as the sole sensor for cloud detection in that region."
* * *
*Authors revised text:* "Insect contamination may contribute to significant uncertainty of the radar-based retrievals of cloud boundaries. Therefore, our analysis employs a semi-quantitative threshold approach when using the cloud top heights. This approach is less sensitive to the insect contamination."

*New reviewers' comment:* Can you clarify what you mean by "semi-quantitative threshold".
* * *
*New reviewer's comment:* Line 463, please consider changing "see" for "detects" without quotation marks.
* * *
*Reviewer's original comment* 7) The idea of compensating bias introduced on Page 8 "introduction of compensating errors using the cloud top height criteria in the updated merged lidar-radar product." needs clarification. - If I understand correctly the hypothesis is that in the 2000-2010 period the MPL was overly sensitive to aerosols leading to a CF overestimation while the MMCR was underly sensitive to cloud leading to a CF underestimation hence the compensating bias

*Authors original response:* Thank you for pointing this out. For the later sub-period (2010-2017), the merged cloud radar-lidar product relies on the "shallow" (< 3 km) radar data instead of the combined MPL-radar observations for determining the cloud top. The radar misses a substantial fraction (about 30%) of ShCu, therefore the cloud top height (below 3 km) is very likely to be missed. Meanwhile, the merged lidars (ceilometer and MPL) data are used to detect the cloud base height and exhibit higher CF than that from the ceilometer alone. A compensating error could potentially arise from the over-detection of clouds in the merged lidar data with the under-detection of cloud from the radar observations. The RMSD for the CF including cloud top heights for the later sub-period (2010-2017) is higher than those for the CF obtained from ceilometer alone (even for near-zero bias). This indicates that the instrument detection differences in the merged lidar-radar product contribute mostly to the CF uncertainty.

*Reviewer's new comment:* Thank for you for the clarification. How has it been incorporated in the revised manuscript?

---

## Author Response (AR2)

More details about the impact of insects on the cloud cover estimates presented should be given; Previous studies have demonstrated that shallow continental cumulus are challenging to detect using radars because of their small numerous droplets. However, when insects are present above cloud base height they can be detected and misinterpreted as cloud if measurements such as LDR are not used for filtering. If cloud detection by radar is a criteria for cloud cover estimate (like it is in the current study), then cloud cover estimate may become biased by the presence/absence of insects which may vary over the course of the year or with temperature.

Thank you for this feedback, and your careful review of our manuscript. Your comments have strengthened this paper throughout the review process. We have adjusted the manuscript to respond to the reviewer's specific comments as described below:

*New reviewers' comment:*

One of the main conclusions of this article is "Whereas CF from merged MPL-ceilometer data provides the largest estimates of the multi-year mean cloud cover: about 0.12 (35%) and 0.08 (24%) greater than FSC for the first and second sub-periods, respectively. CF from merged ceilometer-MPL-radar data has the strongest sub-period dependence with a bias of 0.08 (24%) compared to FSC for the first sub-period and shows no bias for the second sub-period. The strong period dependence of CF obtained from the combined ceilometer-MPL-radar data is likely due to increased reliance on the radar for cloud top height returns."

This statement leaves me with the following thoughts:

1) If the largest factor affect the ceilometer-MPL-radar CF estimate is whether or not the radar detects a cloud top then I think the authors should reconsider their answer to my original comment about the role of insects.

*Reviewers original comment:* "Clarifications regarding the impact of insects on ShCu top detections - Multiple studies have reported that the presence of insect hinders the radars ability to accurately detect cloud top. I think more information is needed here about how insect contamination is handled in ARSCL both pre and post 2011 where the authors hint that the MPL stopped being applied in the boundary layer. This could offer an alternative explanation to the changes in radar-lidar CF post 2011 where the increase in radar detected cloud top could be due both to the KAZR being more sensitive than the MMCR and to the KAZR insect filtering having changed such that more insect returns are misclassified as cloud tops. If both effects are in play, then I would like to see their relative importance quantified."

Let me be clearer about this statement; By "accurately" I meant that sometimes the radar cannot detect the "real" cloud at all but can detect an insect layer which happens to fly above a lidar- detected cloud base. This would lead, for the same "real" cloud, to CF only being estimated by the ceilometer-MPL-radar algorithm if insects are detected and interpreted as being cloud; thus generating biases if the insect filter is altered.

*Authors original response:* "We agree that insects can contaminate accurate determination of cloud boundaries by radar. However, accurate cloud top height retrievals by radar is not required in our analysis because a simple threshold is used to determine the presence of ShCu. Moreover, cloud base height estimation involves lidar observations (both ceilometer and MPL) which are not impacted by the presence of insects, Text has been added to clarify this point: (Section 3.1, lines 141-144) "Insect contamination may contribute to significant uncertainty of the radar-based retrievals of cloud boundaries. Therefore, our

analysis employs a semi-quantitative threshold approach when using the cloud top heights. This approach is less sensitive to the insect contamination."

I agree with the authors that "cloud base height estimation involves lidar observations (both ceilometer and MPL) which are not impacted by the presence of insects". But the importance of detecting "true" cloud tops should not be undermined since it is a fundamental criterion in the ceilometer-MPL-radar CF estimate. Given this, I think the authors should specify how insect filtering is performed both pre and post 2011. I understand that this may have been the topic of previous studies but it would make this manuscript much more thorough to statement the main points of these algorithms; For instance, are they relying on polarimetric variables, are they threshold based, do they have continuity arguments? I would think something has changed after the much improved KAZR was installed post-2011. I also think the potential bias insects could generate should be specifically stated in the conclusions perhaps at the end of point 1).

Response to reviewer: This is an interesting point. While it is possible that insect layers are detected rather than hydrometeors in shallow cumulus cases, there hasn't been a significant change in the algorithm used to detect insects between the ARSCL and the KAZRARSCL period. In private communication with Karen Johnson, she indicated that both periods rely primarily on lidar-based cloud detections to identify insect only events and don't differentiate cloud from insect returns above cloud base. We added the following text to section 3.1 to better clarify that we don't expect insect-clutter to have a significant impact on our cloud fraction results:

Sect 3.1 Line 142

"Insect contamination may contribute to significant uncertainty of the radar-based retrievals of cloud boundaries, however, we do not expect this to significantly impact our results for several reasons. First, when using cloud top height in cloud fraction, we still require cloud base to be identified by the ceilometer or MPL, so we will not misclassify insect-only layers as cloud. Second, our results are not very sensitive to the actual value of cloud top height as long as it is below 4 km, and as most insects will be found in the boundary layer or immediately above it (Kollias et al. 2016; Wainwright et al. 2017) they are not likely to cause the radar to misidentify a cloud height above 4 km if a cloud doesn't exist at that height. Finally, both the ARSCL and KAZRARSCL products rely primarily on lidar-based cloud detections in identifying insect-only events, so we don't expect any systematic differences in the cloud fraction determined by the two products as a result of insects."

Kollias, P., Clothiaux, E. E., Ackerman, T. P., Albrecht, B. A., Widener, K. B., Moran, K. P., Luke, E. P., Johnson, K. L., Bharadwaj, N., Mead, J. B., Miller, M. A., Verlinde, J., Marchand, R. T. and Mace, G. G.: Development and Applications of ARM Millimeter-Wavelength Cloud Radars, Meteorological Monographs, 57, 17.1-17.19, doi:10.1175/AMSMONOGRAPHS-D-15-0037.1, 2016.

Wainwright, Charlotte E et al. "The movement of small insects in the convective boundary layer: linking patterns to processes." Scientific reports vol. 7,1 5438. 14 Jul. 2017, doi:10.1038/s41598-017-04503-0.

2) What does "increased reliance" mean. Going to the conclusion section I get a sense that "increased reliance" means only relying on the radar for cloud top detection below 3km. I would suggest taking the following statement out of the parenthesis in the conclusions "(MPL is not used below 3km in the second sub-period)" since it is a very important part of the sentence. Also, I would suggest making a small change to the abstract to improve clarity: "The strong period dependence of CF obtained from the combined ceilometer-MPL-radar data likely results from a change in what sensors are relied on the detect clouds below 3km; Post 2011, the MPL stopped being used for cloud detection below 3km leaving the radar as the sole sensor for cloud detection in that region."

Response to reviewer: You understood that point correctly. Thank you for your suggestion for improved readability. We have made the corresponding change in the conclusion and the abstract was revised as suggested here:

Abstract line 26: "The strong period dependence of CF obtained from the combined ceilometer-MPL-radar data is likely results from a change in what sensors are relied on to detect clouds below 3 km. After 2011, the MPL stopped being used for cloud top height detection below 3 km, leaving the radar as the only sensor used in cloud top height retrievals"

*Authors revised text:* "Insect contamination may contribute to significant uncertainty of the radar-based retrievals of cloud boundaries. Therefore, our analysis employs a semi-quantitative threshold approach when using the cloud top heights. This approach is less sensitive to the insect contamination."

*New reviewers' comment:* Can you clarify what you mean by "semi-quantitative threshold".

Response to reviewer: We agree that this wording was confusing so we removed that sentence and changed this section of the text to:

Sect 3.1 Line 142

"Insect contamination may contribute to significant uncertainty of the radar-based retrievals of cloud boundaries, however, we do not expect this to significantly impact our results for several reasons. First, when using cloud top height in cloud fraction, we still require cloud base to be identified by the ceilometer or MPL, so we will not misclassify insect-only layers as cloud. Second, our results are not very sensitive to the actual value of cloud top height as long as it is below 4 km, and as most insects will be found in the boundary layer or immediately above it (Kollias et al. 2016; Wainwright et al. 2017) they are not likely to cause the radar to misidentify a cloud height above 4 km if a cloud doesn't exist at that height. Finally, both the ARSCL and KAZRARSCL products rely primarily on lidar-based cloud detections in identifying insect-only events, so we don't expect any systematic differences in the cloud fraction determined by the two products as a result of insects."

*New reviewer's comment:* Line 463, please consider changing "see" for "detects" without quotation marks.

Response to reviewer: Thank you for this suggestion. This change was made.

*Reviewer's original comment* 7) The idea of compensating bias introduced on Page 8 "introduction of compensating errors using the cloud top height criteria in the updated merged lidar-radar product." needs clarification. - If I understand correctly the hypothesis is that in the 2000-2010 period the MPL was overly sensitive to aerosols leading to a CF overestimation while the MMCR was underly sensitive to cloud leading to a CF underestimation hence the compensating bias

*Authors original response:* Thank you for pointing this out. For the later sub-period (2010-2017), the merged cloud radar-lidar product relies on the "shallow" (< 3 km) radar data instead of the combined MPL-radar observations for determining the cloud top. The radar misses a substantial fraction (about 30%) of ShCu, therefore the cloud top height (below 3 km) is very likely to be missed. Meanwhile, the merged lidars (ceilometer and MPL) data are used to detect the cloud base height and exhibit higher CF than that from the ceilometer alone. A compensating error could potentially arise from the over-detection of clouds in the merged lidar data with the under-detection of cloud from the radar observations. The RMSD for the CF including cloud top heights for the later sub-period (2010-2017) is higher than those for the CF obtained from ceilometer alone (even for near-zero bias). This indicates that the instrument detection differences in the merged lidar-radar product contribute mostly to the CF uncertainty.

*Reviewer's new comment:* Thank for you for the clarification. How has it been incorporated in the revised manuscript?

Response to reviewer: The sentence beginning at line 241 (sect 4.1) was changed in this version to:

[revised manuscript text omitted]